# How appraisals of an in-group's collective history shape collective identity and action: Evidence in relation to African identity

Damilola Makanju[1,2]*, Andrew G. Livingstone[1], Joseph Sweetman[1], Chiedozie O. Okafor[3‡], Franca Attoh[4‡]

1 Department of Psychology, Faculty of Health and Life Sciences, University of Exeter, Exeter, United Kingdom, 2 Department of Psychology, School of Social Sciences and Professions, London Metropolitan University, London, United Kingdom, 3 Department of Psychology, Faculty of Social Sciences, Alex Ekwueme Federal University, Achoro-Ndiagu, Ebonyi State, Nigeria, 4 Department of Sociology, Faculty of Social Sciences, University of Lagos, Lagos, Lagos State, Nigeria

☯ These authors contributed equally to this work.
‡ COO and FA also contributed equally to this work.
* dm568@exeter.ac.uk

**Data Availability Statement:** All data and research material files are available from the Open Science Framework database (https://osf.io/qm8g5/?view_only=35292d860cb94550a52d42b01733f87e).

## Abstract

This research tested the impact of how group members appraise their collective history on in-group identification and group-based action in the African context. Across three experiments ($N$s = 950; 270; and 259) with Nigerian participants, we tested whether the effect of historical representations–specifically the valence of the in-group's collective history–on in-group engagement, in turn, depends on whether that history is also appraised as subjectively important. In Study 1, findings from exploratory moderated-mediation analyses indicated that the appraised negative valence of African history was associated with an increase in identification and group-based action when African history was appraised as unimportant (history-as-contrast). Conversely, the appraised positive valence of African history was also associated with an increase in identification and group-based action when African history was also appraised as important (history-as-inspiration). Studies 2a and 2b then orthogonally manipulated the valence and subjective importance of African history. However, findings from Studies 2a and 2b did not replicate those of Study 1. Altogether, our findings suggest that the relationship between historical representations of groups and in-group identification and group-based action in the present is more complex than previously acknowledged.

## Introduction

The way in which group members perceive their in-group's collective history has been shown in social psychological research to shape how group members engage with their in-group. This includes how much they identify with the in-group [e.g., 1, 2], and their willingness to act or mobilise on behalf of the group to achieve its goals [e.g., 3, 4]. However, previous research has primarily focused on appraisals of the valence of collective history–i.e., its overall positivity or

**Funding:** The author(s) received no specific funding for this work.

**Competing interests:** The authors have declared that no competing interests exist.

negativity. Focusing on African identity, in this research we tested the multi-dimensional nature of how group members appraise their in-group's collective history and the impact of those appraisals on in-group identification and group-based action. African identity is defined in our research as part of the self-concept of individuals who identify their origin, cultural roots and/or homeland as stemming from the geographic region of the continent of Africa, and as such can self-categorise themselves as being a member of the African category [5] [p. 2]. Below we review previous work on the impact of historical representations on in-group identification and group-based action, before considering more complex ways in which group members appraise their in-group's collective history and testing the impact of those appraisals on in-group engagement in the case of African identity.

## Historical representations and social identities

History presents a group with narratives of where we came from, who we are, and where we should be going to [6–8]. Narratives of a collective history inform group members' understanding of their social identity and political goals, preferences for ameliorating present challenges facing the group, and conduct in intragroup and intergroup relations [1, 7–12], and helps to delineate a group's customs, traditions, norms, values, symbols, and ideologies for understanding and interacting with the world around them [6, 7, 13].

**Historical representations and in-group identification.**  One important impact of historical narratives is on the relationship that group members have with their in-group. Different representations of history are related to contrasting identification patterns with an in-group [1, 2]. For instance, Licata and colleagues [1] found that in an African sample, perceiving colonialism as exploitative was positively related to national identification, whilst perceiving colonialism as developmental was negatively related to national identification. Correspondingly, Rimé and colleagues [2] found that differences in representations of history between generations of linguistic groups predicted Dutch- and French-speaking Belgians' level of identification with Belgium and their linguistic community. For example, older (in comparison to younger) Dutch-speakers had a collective memory of past victimisation by French-speakers when considering the relationship between Dutch- and French-speakers in Belgium, which then predicted older Dutch-speakers higher identification with their regional category in comparison to the superordinate Belgian category (that includes French-speaking Belgians). Conversely, younger Dutch-speakers identified more with the superordinate Belgian category in comparison to their regional category because their collective memory did not contain past victimisation by French-speakers [2]. Furthermore, more broadly, the perception that an in-group's identity stretches back through time as an enduring entity–that is, perceived collective continuity–is associated with increased in-group identification [14, 15].

**Historical representations and group-based action.**  History is also a resource that is used to shape how a group's collective goals and actions. For instance, historical narratives shape preferences for group-based action aimed at achieving and/or maintaining a positively distinct social identity for the group. More precisely, in Cinnirella's [9, 16, 17] research, findings suggested that European integration was a threat to British identity because of Britain's past colonial strength and domination of world affairs. Therefore, rejecting European identity protected British identity from being eroded by 'Europeanness'. More generally, group leaders utilise historical narratives to define the in-group's identity in specific ways to mobilise group members to fulfil particular political goals [3, 4, 18–21]. For example, Reicher and Hopkins' [4] analysis of Scottish politicians' rhetoric indicates that they evoked narratives of Scottish history in disparate ways to define 'Scottishness' to mobilise Scottish people to either remain part of the United Kingdom or seek independence. Specifically, politicians in favour of the union

with Britain used King Robert the Bruce's victory over England at Bannockburn in 1314 to reject notions that Scotland had a bad deal in the union and to emphasise Scotland's strength, confidence, equality, and success. Conversely, politicians who desired independence used King Robert the Bruce's victory at Bannockburn to emphasise Scotland's life-long battle with England in order to inspire the fight for independence [4].

## Appraising the in-group's collective history

Taken together, the literature reviewed above suggests that representations of an in-group's history shape (1) group members' identification with their in-group, and (2) options for group-based action towards achieving group goals. However, this previous research on how 'ordinary' group members appraise their in-group's collective history has primarily focused on the role of appraisals of *valence*–that is, the overall positivity and negativity–of the in-group's history in in-group identification and group-based action [1, 5, 22–24]. Moreover, findings from these studies reveal contradictions in how appraisals of the valence of the in-group's history shape group members' engagement with the in-group. On the one hand, appraising the in-group's history as negative can hamper group members' collective action tendencies. For example, Rabinovich and Morton [22] found that appraising the in-group's history as negative (as opposed to positive) led to weaker collective action intentions. On the other hand, appraising the in-group's history as negative can mobilise group members towards collective action. For example, Licata and colleagues [1] found that Africans who appraised colonialism as exploitative (as opposed to developmental) predicted more willingness to seek reparations for colonial violence. Other findings also suggest that the connection between appraisals of the in-group's collective history and group members' identification and collective action willingness may not be straightforward: Makanju et al. [5] found that appraisals of positive versus negative African history had no significant effect on African participants' collective political action willingness or their identification as Africans.

These inconsistent findings suggest the value of a more nuanced and multidimensional approach to understanding the impact of appraisals of collective history. To this end, through a narrative literature review of past cross-disciplinary research and literature on the significance of an in-group's collective history to group membership, we delineated thirteen possible dimensions (further clustered under four superordinate-dimensions) along which an in-group's history may be appraised. These dimensions include: (1) s*ubjective importance*, which involves appraising the relevance of the in-group's history to the self (importance to self dimension; e.g., [25]), to the in-group (importance to group dimension; e.g., [7]), to present-day circumstances (importance to present-day dimension; e.g., [6]) and to the world (importance to world dimension; e.g., [26]); (2) *richness*, which involves appraising the in-group's history in terms of how far back in time it stretches (temporality dimension; e.g., [14]), the amount of detail and events (depth dimension; e.g., [27]), and if noteworthy history exists (existence dimension; e.g., [28]); (3) *clarity*, which involves appraising the extent to which the in-group's history is easy-to-understand (comprehensibility dimension; e.g., [29]), consensual (contentiousness dimension; e.g., [4]), and vivid (vividness dimension; e.g., [30]) vs. hard-to-understand, contested, and vague; and (4) *valence*, which involves appraising the extent to which the in-group's history is positive (positivity dimension; e.g., [24]) and emotionally pleasant (pleasantness dimension; e.g., [30]), along with the in-group's welfare through history in terms of glory or suffering (glory dimension; e.g., [9]).

The implied factor structure above (i.e., four superordinate dimensions consisting of subdimensions) is speculative as this is the first quantitative examination of these collective history appraisal dimensions. Therefore, one of the aims of the present research was to examine the

factor structure of these dimensions, as part of our overall aim of quantitatively examining the relationship between these collective history appraisals and in-group engagement.

## The impact of collective history appraisals on in-group engagement: The case of "African" identity

"We do have our own hearts, our own heads, our own history. It is this history which the colonialists have taken from us. The colonialists usually say that it was they who brought us into history: today we show that this is not so. They made us leave history, our history, to follow them, right at the back, to follow the progress of their history" [31] [p. 1].

"The basis of national liberation, whatever the formulas adopted on the level of international law, is the inalienable right of every people to have its own history, and the objective of national liberation is to regain this right usurped by imperialism" [32] [p. 9].

The quotes above from Amilcar Cabral emphasise the damaging effect of colonialism on colonised people's sense of collective history, and also that emancipation from the legacy of colonialism has involved asserting the indigenous/native history of colonised people. Together, they highlight how the legacy of colonialism fundamentally undermines a sense of meaningful collective history.

European colonisation of Africa has shaped the continent immensely, not least of all in determining the current geographic borders and jurisdictions of its sovereign nation-states. Representations of colonialism both in the Global North and Africa have largely depicted it as good, at least on balance [33, 34]. Specifically, colonialist representations of Africa were of a continent that was void of civilisation and 'dark' [35] and that the role of Europeans through benevolence was to rescue, develop, and civilise Africa [34]. This colonialist representation can negatively shape the collective self-worth and self-definition of Africans and how Africans imagine and enact a positive future based on the present and past self-determination of Africans [27, 36, 37]. This is exacerbated by the perceived fit between the colonialist representation of Africa and African nations' current 'developing' (and by extension, struggling) status even after gaining independence [see 38]. Indeed, some Africans fondly and sentimentally long for colonial times, and therefore criticise their governments for the deterioration of political stability, health, education, and economic infrastructures after the end of European colonialism [1, 39].

The colonialist representation of Africa sits in stark contrast to the facts of Africa's precolonial history, which is full of advanced civilisations and achievements in all spheres of life before the violence of colonialism [28, 35]. Consequently, narratives of this (lesser-known) prestigious, precolonial African history have the potential to engender appraisals of a positive, clear, rich, and important collective history that contrasts with colonialist narratives of African history. However, most Africans are unaware of Africa's prestigious precolonial history, at least in part because European colonisation imposed a Western-centric view of history in Africa and destroyed evidence of African civilisations in some cases [40, 41]. Indeed, the nature of history that is taught in African schools propagates the 'benevolence' and 'valour' of European colonisers [27] and is thus void of narratives of colonial violence and prestigious precolonial Africa. Through the erosion of the colonised peoples' collective history, colonialism distorts people's awareness of and pride in a meaningful and relevant (indigenous/native) collective history that pre-dates the colonial era. Accordingly, it is paramount to examine the impact of representations of precolonial African history on Africans' engagement with their in-group (i.e., African identity). Ultimately, this makes the African context an ideal context in which to

examine the multi-dimensional nature of appraisals of collective history and their connection to in-group engagement.

## The present research

Our aim in the present research was to investigate the multi-dimensional nature of how group members appraise their in-group's collective history and the impact of those appraisals on in-group identification and group-based action in the African context. We conducted three experimental studies in which we presented different narratives of African history and measured Africans' appraisals of African history and ingroup engagement. Study 1 was an exploratory investigation that tested the effect of the manipulation of historical representations of precolonial Africans on in-group identification and group-based action. Historical representations of African people were operationalised in broad terms to cover a wide spectrum of African history during precolonial times, including narratives of prestigious precolonial Africa (e.g., global citizenry) and precolonial wrongs of Africans (e.g., inhumane practices such as human sacrifices). Therefore, we manipulated historical representation in three levels in Study 1: positive (prestigious) precolonial African history vs. negative precolonial African history vs. a control condition, focusing on unique aspects of African Savannah wildlife.

Moreover, Study 1 also examined the effect of historical representations on collective history appraisals (i.e., subjective importance, richness, clarity and valence). This is the first quantitative examination of these appraisal dimensions of collective history, and as such, we also conducted an exploratory factor analysis on the structure of the dimensions. We reasoned that exploratory factor analysis is more appropriate than a confirmatory factor analysis because the implied factor structure of how the sub-dimensions/dimensions identified in the introduction related to each other was entirely speculative. Therefore, the speculated factor structure was tentative rather than having a firm theoretical basis a priori, so an exploratory factor analysis in the first instance allowed us to be informed by the data as well as previous research, before a more confirmatory test. Our expectation for the effect of historical representation on collective history appraisals was that representations of positive precolonial African history (vs. negative precolonial history and African Savannah) will lead to higher scores on appraisals of African history as subjectively important, rich, clear, and positive (as opposed to subjectively unimportant, limited, complex and negative). More importantly, our expectation for Study 1 was that representations of positive precolonial African history (vs. negative precolonial history and African Savannah) would lead to higher in-group identification and group-based action.

## Study 1: Method

### Ethics statement

Study 1 (eCLESPsy000533 v8.1) was reviewed and approved by the University of Exeter, School of Psychology Ethics Review Board and all participants gave their written informed consent to participate in the studies. Informed consent was derived by participants responding to and endorsing questions at the start of the studies. It was not possible to start any of the studies without informed consent.

### Participants

The participants were 950 Nigerian adults who all lived in Nigeria. A sensitivity analysis using G*power 3.1 indicated that the final sample of 950 provides 80% power ($\alpha = .05$; $df_{num} = 2$) to detect an omnibus effect as small as Cohen's $f^2 = 0.10$ (equivalent to $\eta_p^2$ of .010) in a one-way analysis of variance (ANOVA). The recruitment strategy was to maximise the sample size

given three months for data collection (data were collected between the 3<sup>rd</sup> of July–the 30th of September of 2019). Most participants (*N* = 905; 95.3%) were recruited via an online link through social media platforms such as Facebook and WhatsApp and were not given any incentive for study participation. A further 41 (4.3%) participants were recruited by a research assistant and were paid 500 Naira (Nigerian currency) in the form of mobile top-ups for study participation. Moreover, a further four participants were recruited via an online paid link, meaning that each participant got paid a 500-naira mobile top-up for their participation. Participants were between 16 and 69 years old (*M* = 20.23, *SD* = 5.30). There were 278 males and 303 females, while one identified their gender as 'other' and three preferred to 'rather not say' (a further 365 did not report their gender/were missing data). Moreover, there were no exclusions after reviewing the data.

## Design

This study had a three-condition, between-participants design and was conducted using Qualtrics. The study's independent variable was historical representations of the African people and had three levels: positive precolonial history (prestigious precolonial Africa); negative precolonial history (precolonial wrongs); and a control condition focusing on the African Savannah. The African Savannah may be viewed as a source of pride for Africans because of its unique variety of wildlife [e.g., 42, 43]. Therefore, the African Savannah condition enabled us to compare the positive precolonial history condition with a positive stimulus to control for positive affect confounds driving any effects, and so to infer that it is historical content itself that may impact changes on in-group identification and group-based action. Dependent variables included collective history appraisals, in-group identification (including group-level self-investment and self-definition), and group-based action which included social competition, consciousness-raising and collective political action.

## Materials

All materials as presented to participants can be found on the project OSF site at https://osf.io/qm8g5/?view_only=35292d860cb94550a52d42b01733f87e. Unless otherwise indicated, responses were made on scales from 1 (*strongly disagree*) to 7 (*strongly agree*).

**Historical representations and control.**   The materials for the historical representations were sourced from Africa's Great Civilisations [28] and Lost Kingdoms of Africa [44] video documentaries on precolonial Africa. In terms of the content of the video representations, the historical conditions covered three topic areas of African history. These areas were introduced using subheadings and are described below under each condition. However, in the negative precolonial history condition, the last two aspects of African history were covered under one subheading because they were inextricably intertwined. Both history-focused conditions started with the same picture which had the words 'The History of Africa'. The material for the control condition was sourced from AnimalWised's [45] video on 'Animals of Africa– 10 wild animals from the African savanna'. The duration of the video clips in minutes was 4:49 for positive precolonial history, 5:00 for negative precolonial history, and 4:16 for control (African Savannah).

*Positive precolonial history*: *Prestigious precolonial Africa*. This historical representation video was intended to depict a decolonised version of precolonial African history that portrayed African history in glorious and positive terms by presenting the high achievements of Africans which showed highly civilised peoples before colonialism. The subheadings of this video were: *great scholarship*, *civilised people* and *stunning artistry*.

*Negative precolonial history*: *Precolonial wrongs of Africans*. This historical representation video portrayed precolonial African history in negative terms, focusing on past mistakes and inhuman practices. The subheadings of this video were: *human sacrifices* and *loss of skilled Africans and greed in African-led slave trade*.

*African savannah*. The control condition video was a positive, feel-good depiction of the African Savannah, focusing on ten animals that are exclusive to the continent of Africa. The animals–starting from the African Elephant and ending with the Ring-tailed Lemur–were listed from one to ten in alphabetically ascending order.

**Collective history appraisals.** Items to assess participants' appraisals of Africa's collective history consisted of 13 semantic differential items. Conceptually, these items were developed to assess different dimensions along which an in-group's collective history could be appraised. Examples of the items include: 'stretches back thousands of years–stretches back only a short time' (temporality dimension), 'extremely rich–extremely limited' (depth dimension), 'mainly glorious–mainly of suffering' (glory dimension) and 'important to me–not important to me' (importance to self dimension). The items were prefaced with the statement 'African history is:'. Responses were scored from -3 (*negatively anchored scale end*) to 3 (*positively anchored scale end*). However, the response scale as visible to participants was not numbered to avoid attaching implied value to one type of response.

**Identification.** *Group level self-definition and self-investment*. The 14-item measure (α = .91) of identification from Leach and colleagues [46] was used to assess participants' African identification. For self-definition, there were four items (α = .79; e.g., 'I have a lot in common with the average African person' and 'African people have a lot in common with each other') and for self-investment, there were 10 items (α = .89; e.g., 'I am glad to be African' and 'I feel a bond with Africans').

**Group-based action.** *Social competition*. Four items (α = .85) adapted from Blanz et al., [47] were used to assess participants' endorsement of the extent to which Africa should compete with the West (e.g., 'Africans as a group should try and achieve equality with the West' and 'Africans should try to be better than the West').

*Consciousness-raising*. Five items (α = .86) were developed to assess participants' willingness to raise consciousness amongst Africans about African history (e.g., 'tell other Africans about African history' and 'volunteer for an organisation that informs Africans about African history'), using a scale from 1 (*very unwilling*) to 7 (*very willing*).

*Collective political action*. Nine items (α = .88) adapted from Sweetman and colleagues [48] were used to assess participants' willingness to engage in political actions (e.g., 'help organise a petition', 'donate money to the cause' and 'join a social movement focusing on Africa'), using a scale from 1 (*very unwilling*) to 7 (*very willing*).

**Additional measures.** This study contained some additional measures that were included for exploratory reasons and whose data are not analysed here. These were scales of social-structural perceptions [adapted from 49] which included: stability, legitimacy and permeability of group boundaries; a two-item scale of individual mobility [adapted from 47]; a 12-item measure [adapted from 50] assessing collective self-esteem; and a single-item assessing participants' appraisals of their knowledge of African history.

## Procedure

Participants were informed that the experiment was a survey containing video representations of African history and questionnaires on their opinions of Africa. First, participants were randomly allocated to one of the three treatment conditions and presented with the respective video clips of their assigned condition. After participants saw the video narratives, they

completed the appraisals of collective history scale. Next, participants completed the measures on identification and group-based action, followed by demographic information (i.e., age, place of birth, African nationality and ethnicity, citizenship, gender, and education level). For participants in the negative precolonial history condition, a special debrief was included to curtail any negative reactions from being shown negative information about Africa. The debrief clarified the conceptual stance of the negative precolonial historical representation that participants watched and gave them the historical representations that participants watched in the positive precolonial history condition in textual form (i.e., bullet points). Lastly, participants were thanked and debriefed on the purposes, hypotheses, and expected outcomes of the research.

## Results

### Exploratory factor analysis: Appraisals of collective history scale

A principal axis factor analysis was conducted on the 13 collective history appraisal dimension items with oblique rotation (direct oblimin). Where relevant, items were reverse scored so that the positive end of the scale signified higher scores on all items. The Kaiser-Meyer-Olkin measure verified the sampling adequacy for the analysis, KMO = .88 ['meritorious' according to 51], and all KMO values for individual items were greater than .67, which is well above the acceptable limit of .5 [52]. An initial analysis was run to obtain eigenvalues for each factor in the data. Three factors had eigenvalues over Kaiser's criterion of 1 and in combination explained 64.70% of the variance. The scree plot showed an inflexion that would justify retaining three factors. Three factors were retained because of the large sample size and the convergence of the scree plot and Kaiser's criterion on this value. Table 1 shows the factor loadings after rotation. The items that cluster on the same factor suggest that factor 1 represents the subjective importance of the in-group's history, factor 2 represents the clarity of the in-group's

**Table 1. Summary of exploratory factor analysis results for the appraisals of history scale (_N_ = 584).**

| Item | Rotated Factor Loadings | | |
|---|---|---|---|
| | 1 ('Subjective importance') | 2 ('Valence') | 3 ('Clarity') |
| _Important to the group_: not important to Africans–important to Africans | **.87** | -.10 | .05 |
| _Temporality_: stretches back only a short time–stretches back thousands of years | **.82** | .03 | -.10 |
| _Existence_: something that doesn't exist–something that definitely exists | **.82** | .03 | -.10 |
| _Importance to the world_: not important to the world–important to the world | **.69** | .00 | .04 |
| _Importance to present day_: irrelevant to the present–relevant to the present | **.64** | .11 | .11 |
| _Importance to self_: not important to me–important to me | **.63** | -.06 | .17 |
| _Richness_: extremely limited–extremely rich | **.58** | .23 | -.07 |
| _Glory_: mainly of suffering–mainly glorious | -.04 | **.90** | -.05 |
| _Pleasantness_: unpleasant to think about–pleasant to think about | -.01 | **.73** | .09 |
| _Positivity_: highly negative–highly positive | .15 | **.54** | .12 |
| _Vividness_: vague or unclear (hard to bring to mind)–vivid or clear (easy to bring to mind) | .04 | .01 | **.83** |
| _Comprehensibility_: hard to understand–easy to understand | .07 | .07 | **.58** |
| _Contentiousness_: controversial or people disagree about it–uncontroversial or people agree about it | -.04 | .02 | **.52** |
| Eigenvalues | 5.47 | 1.69 | 1.26 |
| % of variance | 42.04 | 13.01 | 9.65 |
| α | .89 | .80 | .70 |

_Note_: Factor loadings over .40 appear in bold

**Table 2. Bivariate correlations of all variables in Study 1.**

| | 1 | 2 | 3 | 4 | 5 | 6 | 7 |
|---|---|---|---|---|---|---|---|
| 1. IMP | 1 | | | | | | |
| 2. CLA | .38** | 1 | | | | | |
| 3. VAL | .46** | .40** | 1 | | | | |
| 4. SELF-I | .19** | .15** | .13** | 1 | | | |
| 5. SELF-D | .13** | .14** | .10* | .60** | 1 | | |
| 6. SCO | .12** | .11** | .04 | .22** | .23** | 1 | |
| 7. CON-RA | .10** | .17** | .02 | .33** | .16** | .17** | 1 |
| 8. COLACT | .11** | .14** | .06 | .35** | .20** | .19** | .65** |

Notes.

* $p$ a< .05

** $p < .01$

1. IMP, Subjective importance of history; 2. CLA, Clarity of history; 3. VAL, Valence of history; 4. SELF-I, Self-investment; 5. SELF-D, Self-definition; 6. SCO, Social competition; 7. CON-RA, Consciousness-raising; and 8. COLACT, Collective political action

history and factor 3 the valence of the in-group's history. Listwise deletion was used for the exploratory factor analysis. Hence, the reduced N = 584.

Bivariate correlations of the dimensions to the appraisals of history and all variables in the study can be found in Table 2.

## Appraisals of collective history

To check the impact of the manipulation of historical representations on participants' appraisals of African history, a between-participant MANOVA was conducted with historical representation condition (i.e., positive precolonial history, negative precolonial history, and African Savannah) as a three-level factor. This analysis was performed with the three dimensions (i.e., subjective importance, $\alpha = .89$; clarity, $\alpha = .70$; and valence, $\alpha = .80$) to the appraisals of collective history derived from the exploratory factor analysis, and the scales were formed by averaging scores on the items on each dimension. Using Wilks' lambda, there was a significant effect of historical representation condition, $\lambda = 0.81$, $F(6, 1156) = 22.09$, $p < .001$, $\eta_p^2 = .102$.

Results from the follow-up analyses revealed a large, significant effect of historical representation on participants' appraisals of the *valence* of African history $F(2, 583) = 48.84$, $p < .001$, $\eta_p^2 = .143$, with participants in the African Savannah condition ($M = 1.23$, $SD = 1.33$) appraising African history as more positively-valenced in comparison to participants in the positive precolonial history ($M = 1.11$, $SD = 1.60$) and negative precolonial history ($M = -0.10$, $SD = 1.45$) conditions. Pairwise comparisons between the two historical representation conditions showed that positive precolonial history led to higher scores (i.e., significant mean difference; $MD$) on appraisals of the valence of African history in comparison to negative precolonial history ($MD = 1.21$, $SE = 0.15$, $p < .001$).

Additionally, there was a small, but significant effect of historical representation on participants' appraisals of the *subjective importance* of African history, $F(2, 583) = 4.70$, $p = .009$, $\eta_p^2 = .016$, with participants in the African Savannah condition ($M = 2.07$, $SD = 1.04$) appraising African history as more important in comparison to participants in the negative precolonial history ($M = 1.85$, $SD = 1.13$) and positive precolonial history ($M = 1.68$, $SD = 1.53$) conditions. However, pairwise comparisons between the two historical representation conditions showed that positive precolonial history did not lead to significantly higher scores on appraisals of the subjective importance of African history in comparison to the negative precolonial history ($MD = -0.17$, $SE = 0.13$, $p = .190$).

**Table 3. Pairwise comparisons of ANOVA effects of historical representation on appraisals of collective history in Study 1.**

| Appraisals of history | Reference | Comparison | Mean difference | SE | p |
|---|---|---|---|---|---|
| Subjective importance | Positive precolonial history | Negative precolonial history | -0.17 | 0.13 | .190 |
| | | African savannah | -0.39 | 0.13 | .002 |
| Clarity | Positive precolonial history | Negative precolonial history | 0.08 | 0.14 | .604 |
| | | African savannah | -0.20 | 0.14 | .167 |
| Valence | Positive precolonial history | Negative precolonial history | 1.21 | 0.15 | < .001 |
| | | African savannah | -0.12 | 0.15 | .416 |

The effect of historical representation on participants' appraisals of the *clarity* of African history was not significant, $F(2,583) = 1.91$, $p = .149$, $\eta_p^2 = .007$, with participants in the African Savannah condition ($M = 0.77$, $SD = 1.33$) having similar scores to participants in the positive precolonial history ($M = 0.57$, $SD = 1.48$) and negative precolonial history ($M = 0.50$, $SD = 1.44$) conditions. Pairwise comparisons for all ANOVAs are shown in Table 3.

Unexpectedly, the ANOVA results thus showed that the African Savannah (i.e., control) condition had the highest scores on all collective history appraisal dimensions. Furthermore, the ANOVA results show that the manipulation of historical representations was much more effective–that is, had a larger magnitude of effect–on appraisals of the valence of African history ($\eta_p^2 = .143$) in comparison to appraisals of subjective importance ($\eta_p^2 = .016$) and clarity ($\eta_p^2 = .007$) of African history. Moreover, the effect of the manipulation on valence was the only result for the collective history appraisals that had a significant mean difference in the direction that was expected regarding the comparison between the two historical representation conditions.

## Identification

A between-participants MANOVA with historical representation condition as a three-level factor was conducted to test the effect of historical representation on the strength of African identification (self-investment and self-definition). Using Wilks' lambda, there was a non-significant effect of historical representation, $\lambda = 0.98$, $F(4, 1186) = 0.42$, $p = .792$, $\eta_p^2 = .001$. Moreover, results from the follow-up analyses from the ANOVAs revealed consistent null effects of historical representation on (1) self-investment, $F(2, 594) = 0.20$, $p = .823$, $\eta_p^2 = .001$; and (2) self-definition, $F(2, 594) = 0.39$, $p = .675$, $\eta_p^2 = .001$. The descriptive statistics are reported by condition for both identification variables in Table 4. All pairwise comparisons of ANOVA effects of historical representations on variables of identification can be found on the project OSF site at https://osf.io/qm8g5/?view_only=35292d860cb94550a52d42b01733f87e.

**Table 4. Descriptive statistics for Study 1's follow-up ANOVA effects on dependent variables (i.e., identification & group-based action).**

| Dependent variable | Historical representation conditions | | | | | |
|---|---|---|---|---|---|---|
| | Positive precolonial | | Negative precolonial | | African Savanah | |
| Identification | M | SD | M | SD | M | SD |
| Self-investment | 5.53 | 1.17 | 5.50 | 1.25 | 5.45 | 1.28 |
| Self-definition | 5.41 | 1.32 | 5.29 | 1.44 | 5.38 | 1.39 |
| | Positive precolonial | | Negative precolonial | | African Savanah | |
| Group-based action | M | SD | M | SD | M | SD |
| Consciousness-raising | 5.51 | 1.44 | 5.77 | 1.31 | 5.42 | 1.54 |
| Social competition | 5.71 | 1.48 | 5.98 | 1.50 | 5.83 | 1.45 |
| Collective political action | 5.22 | 1.25 | 5.42 | 1.22 | 5.12 | 1.44 |

### Group-based action

A similar MANOVA was conducted to test the effect of historical representation on group-based action. Using Wilks' lambda, there was a non-significant effect of historical representation on group-based action, $\lambda = 0.98$, $F(6, 1158) = 1.55$, $p = .157$, $\eta_p^2 = .008$. Results from the follow-up analyses did reveal a small, but just-significant effect of historical representation on participants' willingness to raise consciousness (i.e., raise awareness) about African history $F(2, 581) = 3.13$, $p = .044$, $\eta_p^2 = .011$, with participants in the negative precolonial history condition ($M = 5.77$, $SD = 1.31$) being more willing to raise consciousness about African history in comparison to participants in the positive precolonial history ($M = 5.51$, $SD = 1.44$) and African Savannah ($M = 5.42$, $SD = 1.54$) conditions. Other results from the follow-up analyses from the ANOVAs revealed consistent null effects of treatment condition on (1) social competition, $F(2, 581) = 1.61$, $p = .201$, $\eta_p^2 = .006$; and (2) collective political action, $F(2, 581) = 2.60$, $p = .075$, $\eta_p^2 = .009$. The descriptive statistics are reported by condition for all group-based action variables in Table 4. All pairwise comparisons of ANOVA effects of historical representations on variables of group-based action can be found on the project OSF site at https://osf.io/qm8g5/?view_only=35292d860cb94550a52d42b01733f87e.

### Exploratory analysis: The role of the measured appraisals of collective history in the relationship between historical representations and identification/group-based action

The findings of Study 1 revealed null effects of the manipulation on the dependent variables. Given that the primary aim of the present research was to investigate the multi-dimensional nature of collective history appraisals, we then conducted some exploratory analysis to investigate the multi-dimensional nature of how group members appraise their in-group's collective history and the impact of those appraisals on in-group identification and group-based action. We did this primarily by exploring the impact of the manipulation of historical representations on the measured appraisals of collective history.

The results from this study so far show that the measured appraisals of valence and (to a lesser extent) subjective importance were successfully varied by our manipulation of historical representations of precolonial Africa. Consequently, the focus of our exploratory analysis was on the measurements of these two dimensions. Given the large effect of the manipulation on the measured appraisals of the valence of African history ($\eta_p^2 = .142$), we decided to conceptualise the appraised valence as the mediator of the relationship between historical representation and identification/group-based action–that is, that the manipulation may 'work' indirectly via appraisals of valence. This also means that the manipulation of historical representations should in hindsight be considered primarily as a specific manipulation of valence. Moreover, the very small size of the effect of the manipulation ($\eta_p^2 = .016$) on the measured appraisals of subjective importance (even though significant because of the study's large sample size) suggests that the appraised subjective importance is largely orthogonal to the effect of the manipulation. In view of this, we subsequently examined the measured appraisals of subjective importance as a moderator of the indirect effect of historical representation on identification/group-based action via appraised valence.

The conceptualisation of the appraised valence as a mediator and the appraised subjective importance as a moderator aligns with the notion in appraisal theories of emotion [e.g., 53–55] and attitude theories [e.g., 56–58]: the influence of our valence appraisals is greater if we also appraise the object as important to us.

Altogether, our exploratory analysis tested the conditional indirect effect of historical representation condition on identification and group-based action via the measure of appraised

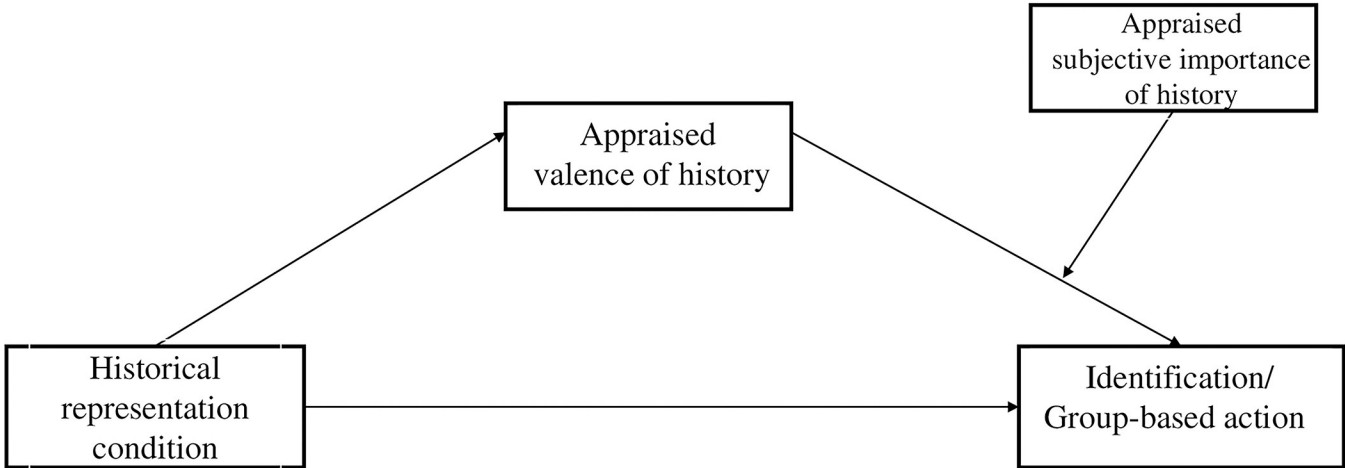

**Fig 1. Conceptual diagram of the conditional indirect effect of historical representation on identification/group-based action through the measure of appraised valence of history at different levels of the measure of appraised subjective importance of history.** The variable of African historical representation condition is a dummy coded variable of the comparison of positive precolonial history (referent category) to negative precolonial history and African Savannah (i.e., control condition).

valence of African history at different degrees (i.e., high and low levels) of the measure of appraised subjective importance of African history (see Fig 1). Our exploratory expectation was that there would be an indirect effect of historical representation of African people on in-group identification and group-based action explained by the appraised valence of African history at different degrees (i.e., high and low levels) of appraised subjective importance of African history.

Below we report the results of our analysis by dependent variable (i.e., identification and group-based action). The variables of group level self-investment and self-definition were averaged to form the identification variable ($\alpha = .91$) because they all assessed the relationship group members have with their social identity/in-group. Moreover, the variables of social competition, consciousness-raising and collective political action were averaged to form the group-based action variable ($\alpha = .90$) because they all assessed group-oriented directions an in-group member may take/hold to achieve progressive goals for the in-group. More importantly, the patterns of results observed from the averaged variables of identification and group-based action in this exploratory analysis were also observed for the individual variables. We report the results of the exploratory analysis for the individual variables of identification and group-based action in the supplementary material on the project OSF site: https://osf.io/qm8g5/?view_only=35292d860cb94550a52d42b01733f87e. The bivariate correlations between the appraisals of valence and subjective importance of African history, identification and group-based action can be found in Table 5.

**Identification.** All the component path coefficients are reported in Fig 2 below. Dummy coding was utilised, with the positive precolonial history condition employed as the reference category for comparisons to the other experimental conditions. Specifically, the positive precolonial history condition was coded as 0 and the other experimental conditions (i.e., negative precolonial history and African Savannah) were coded as 1. Using the PROCESS add-on for SPSS [59], moderated mediation analysis using bootstrapping with 5000 samples indicated that the indirect effect of historical representation on identification via appraised valence of history is contingent on individuals' appraised subjective importance of history. This was only true when the negative precolonial history condition was compared to the positive precolonial

**Table 5. Bivariate correlations between appraisals of collective history, identification, and group-based action for Study 1.**

|  | 1 | 2 | 3 | 4 |
|---|---|---|---|---|
| 1. Subjective importance | 1 |  |  |  |
| 2. Valence | .46** | 1 |  |  |
| 3. Identification | .18** | .13** | 1 |  |
| 4. Group based action | .15** | .06 | .37** | 1 |

Notes.

** $p < .01$

history condition, with the moderated mediation index = -.22, *SE* = .04, *95% CI* = [-.31, -.14]. The indirect effect of the negative precolonial history compared to positive precolonial history on identification via appraised valence of history was significant at low levels of appraised subjective importance of history, with *b* = .21, *SE* = .07, 95% *CI* = [.08, .36]. This means that negative appraisals of African history actually predicted *greater* in-group identification when African history was also appraised as subjectively unimportant and may suggest history being used as a contrast to boost identification. Framed differently, positive appraisals of African history predicted a decrease in in-group identification when African history was also appraised as subjectively unimportant.

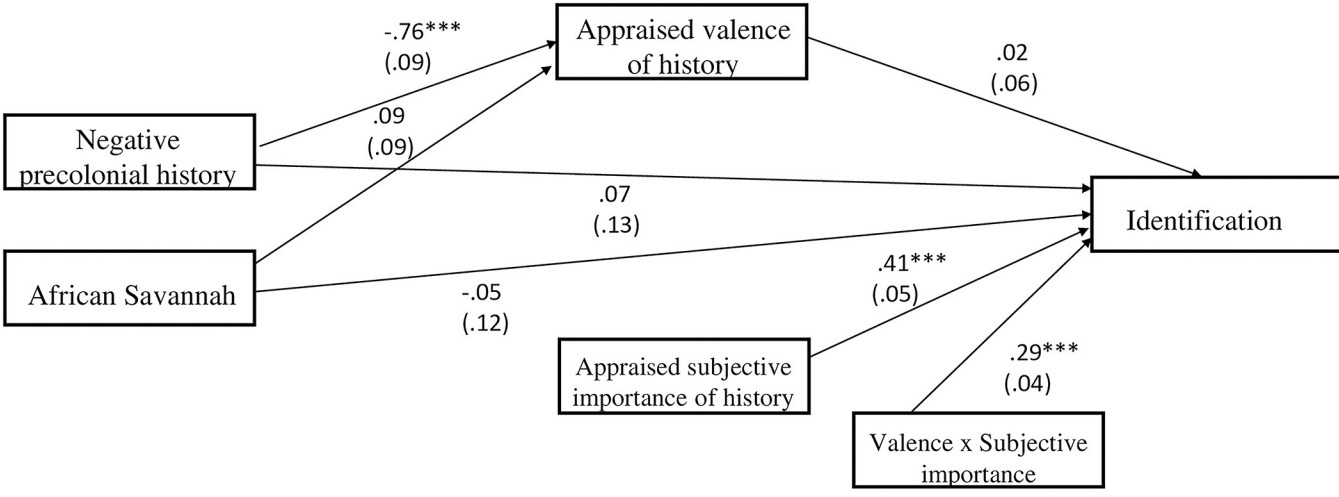

Moderated mediation index for negative precolonial history = -.22, *SE* = .04, *95% CI* = [-.31, -.14]
a*b at low importance, *b* = .21 (.07), 95% *CI* = [.08, .36]
a*b at mean, *b* = -.01 (.05), 95% *CI* = [-.11, .08]
a*b at high importance, *b* = -.21 (.06), 95% *CI* = [-.33, -.11]

Moderated mediation index for African Savannah = .02, *SE* = .03, *95% CI* = [-.03, .08]
a*b at low importance, *b* = -.02 (.03), 95% *CI* = [-.08, .03]
a*b at mean, *b* = .00 (.01), 95% *CI* = [-.01, .02]
a*b at high importance, *b* = .02 (.03), 95% *CI* = [-.03, .08]

**Fig 2. Results of the moderated mediation model of historical representation on identification via the measure of appraised valence of history contingent on the measure of appraised subjective importance of history for Study 1.** Note. The variables of negative precolonial history and African savannah (coded as 1) are dummy variables of their comparison with the positive precolonial history (referent category; coded as 0). The indirect effect is contingent upon the subjective importance of history. Unstandardized regression coefficients are reported with standard errors in parentheses. Path entries are unstandardized coefficients. *** $p < .001$.

In contrast, when appraised subjective importance of history was at high levels, the indirect effect of the negative precolonial history compared to positive precolonial history on identification via appraised valence of history was also significant, but negative, *b* = -.21, *SE* = .06, 95% *CI* = [-.33, -.11]. This means that positive appraisals of African history predicted *greater* in-group identification when the history was also appraised as subjectively important. This may suggest history being used as an inspiration to boost identification.

Furthermore, the indirect effect of the negative precolonial history compared to positive precolonial history on identification via appraised valence of history was not significant for the mean level of appraised subjective importance of history, with *b* = -.01, *SE* = .05, 95% *CI* = [-.11, .08].

**Group-based action.** All the component path coefficients are reported in Fig 3 below. Again, dummy coding was utilised, with the positive precolonial history condition selected as the reference category for comparisons to other experimental conditions. Specifically, the positive precolonial history condition was coded as 0 and the other experimental conditions (i.e., negative precolonial history and African Savannah) were coded as 1. A similar moderated mediation analysis revealed that the indirect effect of historical representation on group-based action via appraised valence of history was contingent on individuals' appraised subjective importance of history. This was only true when the negative precolonial history condition was compared to the positive precolonial history condition, with the moderated mediation index =

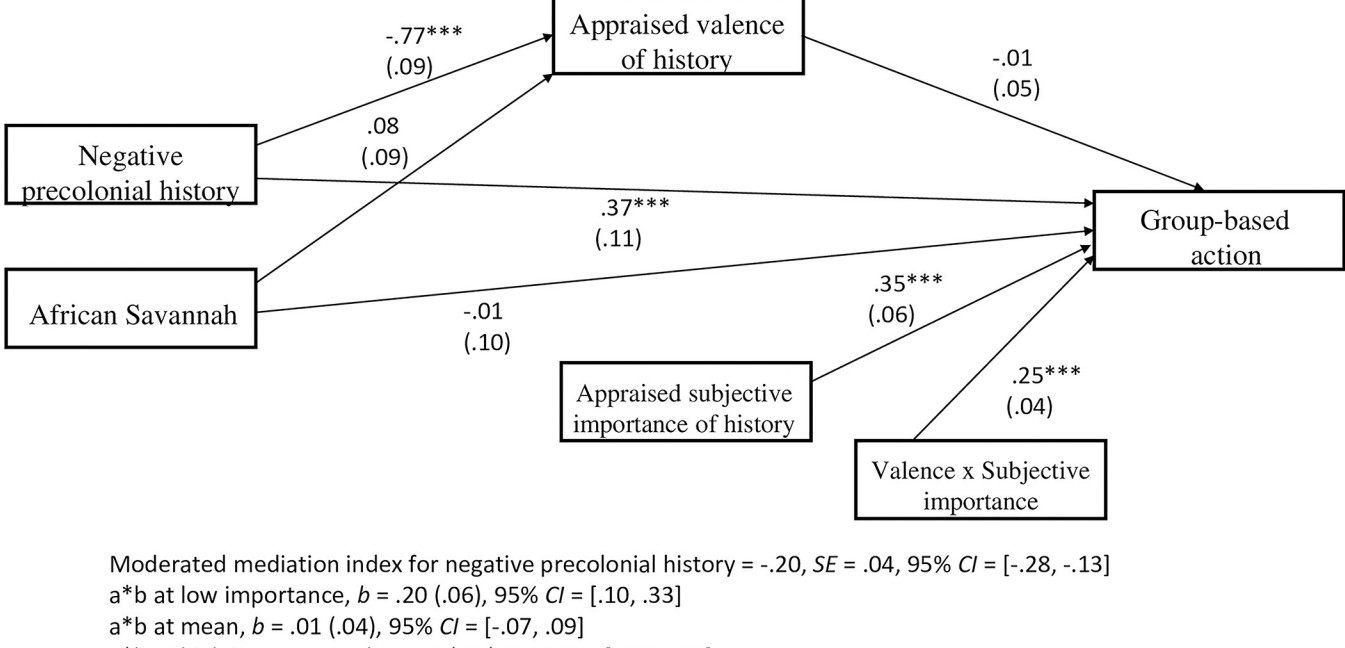

Moderated mediation index for negative precolonial history = -.20, *SE* = .04, 95% *CI* = [-.28, -.13]
a*b at low importance, *b* = .20 (.06), 95% *CI* = [.10, .33]
a*b at mean, *b* = .01 (.04), 95% *CI* = [-.07, .09]
a*b at high importance, *b* = -.17 (.05), 95% *CI* = [-.27, -.08]

Moderated mediation index for African Savannah = .02, *SE* = .02, *95% CI* = [-.03, .07]
a*b at low importance, *b* = -.02 (.03), 95% *CI* = [-.08, .03]
a*b at mean, *b* = -.00 (.01), 95% *CI* = [-.02, .01]
a*b at high importance, *b* = .02 (.02), 95% *CI* = [-.02, .07]

**Fig 3. Results of the moderated mediation model of historical representation on group-based action via the measure of appraised valence of history contingent on the measure of appraised subjective importance of history for Study 1.** Note. The variables of negative precolonial history and African savannah (coded as 1) are dummy variables of their comparison with the positive precolonial history (referent category; coded as 0). The indirect effect is contingent upon the subjective importance of history. Unstandardized regression coefficients are reported with standard errors in parentheses. Path entries are unstandardized coefficients. *** *p* < .001.

-.20, *SE* = .04, 95% *CI* = [-.28, -.13]. The indirect effect of the negative precolonial history compared to positive precolonial history on group-based action via appraised valence of history was significant for low levels of appraised subjective importance of history, *b* = .21, *SE* = .06, 95% *CI* = [.10, .34]. This means that negative appraisals of African history actually predicted *greater* group-based action when African history was also appraised as subjectively unimportant and may suggest history being used as a contrast to boost group-based action. Framed differently, positive appraisals of African history actually predicted lower group-based action when African history was also appraised as subjectively unimportant.

In contrast, when appraised subjective importance of history was at high levels, the indirect effect of the negative precolonial history compared to positive precolonial history on group-based action via appraised valence of history was also significant, but of the opposite sign, *b* = -.17, *SE* = .05, 95% *CI* = [-.27, -.08]. This means that positive appraisals of African history actually predicted *greater* group-based action when African history was also appraised as subjectively important, again suggestive of history being used as an inspiration to boost group-based action. Furthermore, the indirect effect of the negative precolonial history compared to positive precolonial history on group-based action via appraised valence of history was not significant for the mean level of appraised subjective importance of history, with *b* = .01, *SE* = .04, 95% *CI* = [-.07, .09].

## Discussion

The primary aim of this study was to examine the impact of historical representations of African people on in-group identification and group-based action, and the role of collective history appraisals in these links. We did so by testing the effect of positive vs. negative historical representations of precolonial Africans on (1) appraisals of African history as subjectively important, rich, clear, and positive, and (2) in-group engagement in the form of in-group identification and group-based action.

Results of the exploratory factor analysis on the appraisals of collective history items revealed a three-factor structure in which dimensions of positivity, glory and pleasantness indicate the *valence* of the in-group's history; the dimensions of depth, temporality, existence, importance to the self, group, present-day and world indicate the *subjective importance* of the in-group's history; and the dimensions of comprehensibility, contentiousness and vividness indicate the dimensions of the *clarity* of the in-group's history. This factor structure differs from that delineated in the introduction where the dimensions of depth, temporality and existence formed a separate (and fourth) superordinate dimension of the richness of the in-group's history. Altogether, this indicates that the dimensions of depth, temporality and existence signify a well-established collective history and may therefore contribute to the appraisals of how subjectively important history is to an in-group or group members.

Moreover, results indicated that the manipulation of historical representations only showed a large, significant effect on appraised subjective valence and a small effect on appraised importance. This result suggested that the historical representation manipulations primarily varied the valence of African history rather than any other dimension of collective history appraisals. In other words, the videos on precolonial African history were only effective in varying how participants appraised the valence of African history.

Unexpectedly, participants in the control (African Savannah) condition reported higher scores on appraised subjective importance and valence of African history than participants in the positive precolonial history and negative precolonial history conditions. This may point to the positivity and pride obtained from these unique habitats, which then boosted ratings on the collective history appraisal dimensions. However, it is also possible that the effect of the

African Savannah condition may instead point to reactance [e.g., 60] in defence of African history and identity from the perceived stereotype of being animalist which was a popular racist and colonialist representation of Africans [61].

In terms of effects on outcomes related to in-group engagement, the results provided no support for our expectation that positive precolonial history (vs. negative precolonial history and African Savannah) would lead to higher African identification and group-based action. Given the primary aim of this research was to investigate the multi-dimensional nature of appraisals of collective history, we subsequently decided to explore the conditional indirect effect of historical representation on in-group identification and group-based action via (measured) appraised valence of history at different degrees (i.e., high and low levels) of (measured) appraised subjective importance history (see Fig 1). This exploratory analysis was informed by our data (i.e., the results of the manipulation of historical representations on appraised valence and subjective importance of African history). Therefore, we examined the possibility that the effect of historical representation of African people on in-group identification and group-based action would be explained by the appraised valence of African history at different degrees (i.e., high and low levels) of appraised subjective importance of African history.

Results from the exploratory analysis showed that historical representations had a clear effect on how positively or negatively African history was appraised to be. This appraised valence in turn predicted African identification and group-based action differently depending on the appraised subjective importance of African history. Specifically, for participants in the positive precolonial history condition compared to participants in the negative precolonial history condition, appraising African history as more positive predicted greater African identification and group-based action when that history was also appraised as subjectively important; which may suggest a possible history-as-inspiration strategy to boost identification and group-based action.

In contrast, appraising African history as more negative also predicted greater African identification and group-based action when African history was also appraised as subjectively *unimportant*; which may suggest a possible history-as-contrast strategy to boost identification and group-based action.

The possible history-as-inspiration and history-as-contrast patterns found in the exploratory analysis are similar to the findings of Makanju et al.'s [62] qualitative work on the role of collective history appraisals in in-group engagement of Africans. Specifically, Makanju and colleagues [62] found that appraisals of African history as positive and subjectively important were accompanied by history being characterised as an inspiration–to gain impetus from Africa's past–to boost positive engagement with African identity. This strategy of history-as-inspiration to boost in-group engagement is common in previous social psychology research [e.g., 4, 6, 7, 9, 13, 22, 24, 30]. Conversely, Makanju and colleagues [62] found that appraisals of African history as negative and subjectively unimportant to the self-concept were accompanied by history being deployed as a contrast–to break away from Africa's past–to boost positive engagement with African identity. This strategy of history-as-contrast in achieving group goals also echoes the findings of Klein et al., [63]. Specifically, Klein and colleagues studying the Belgian linguistic conflict found that the collective past between the then low-status Dutch-speaking and high-status French-speaking Belgians was used as a contrast to present relations between the now high-status Dutch-speakers and low-status French-speakers to legitimise calls for autonomy by Dutch-speakers vs. interdependence by French-speakers.

At the same time, we are very cautious regarding causal inference from these findings, not only because they were exploratory, but also because the role of appraised valence and subjective importance of collective history were established from cross-sectional data. In an attempt to address this limitation, Studies 2a and 2b aimed to replicate the findings of the exploratory

analysis of Study 1 by testing the causal role of collective history appraisals of valence and subjective importance on in-group identification and group-based action. In addition to manipulating the valence of African history, Studies 2a and 2b tested a manipulation of the subjective importance of collective history through varying/framing descriptive social norms around how Africans appraise the importance of African history.

## Studies 2a and 2b

We amended the design used in Study 1 in several ways. First, we included a manipulation of the subjective importance of collective history, achieved by framing descriptive norms [e.g., 64–66] around how other Africans appraised the importance of African history through fictitious new articles. This also involved using a *three-reason manipulation* [67] to reinforce and increase participants' engagement with the manipulation stimuli on the subjective importance of African history.

Second, we removed the African Savannah control condition because of possible reactance to this material as discussed above, and the fact that it was not a meaningful comparison to the historical representation conditions in the exploratory findings of possible history-as-inspiration and history-as-contrast patterns for African identification and group-based action.

The two other operational changes included (1) changing the measures of appraisals of collective history from semantic differential to an attitude statement format (which is the normative format of most measures in social psychology). This approach had the additional advantage of employing items that have been subjected to confirmatory factor analysis that confirmed the three-factor structure of the exploratory factor analysis results obtained in Study 1 (see confirmatory factor analysis and fit indices in the document 'CFA for appraisals of collective history scale Studies 2a & 2b' on the project OSF site at: https://osf.io/qm8g5/?view_only=35292d860cb94550a52d42b01733f87e), and (2) changing the identification scale to a four-item measure to reduce the number of items in the questionnaire.

Due to unexpected complications in data collection for Study 2a, we conducted Study 2b as a direct replication of Study 2a to ensure the reliability of the results. Specifically, data for Study 2a was collected individually wherever participants felt most comfortable completing the online survey, whereas data for Study 2b was collected collectively in a lecture hall over two days. In other words, participants in Study 2b were invited to a lecture hall to complete the online survey with other participants in the same room. Therefore, since we could not establish uniformity in the procedures of Studies 2a and 2b, we decided to analyse these studies separately.

Studies 2a and 2b employed a 2 (subjective importance of African history: important history norm vs. unimportant history norm) X 2 (historical representation: positive precolonial African history vs. negative precolonial African history) study design. In line with the exploratory findings from Study 1, we predicted that the effect of historical representations–specifically the valence of African history–will also depend on whether African history is appraised as subjectively important, such that there will be an interaction of historical representations and subjective importance of African history. Specifically, for participants in the positive precolonial African history condition, the exposure to the condition of subjective importance of African history was expected to lead to higher in-group identification and group-based action in comparison to the condition of subjective unimportance of African history (history-as-inspiration prediction); and for participants in the negative precolonial African history condition, the exposure to the condition of subjective unimportance of African history was also expected to lead to higher in-group identification and group-based action in comparison to the subjective importance condition (history-as-contrast prediction).

## Method

### Ethics statement

Studies 2a and 2b (eCLESPsy001589 v3.1, eCLESPsy001589 v7.1 and eCLESPsy001589 v9.1) were reviewed and approved by the University of Exeter, School of Psychology Ethics Review Board and all participants gave their written informed consent to participate in the studies. Informed consent was derived by participants responding to and endorsing questions at the start of the studies. It was not possible to start any of the studies without informed consent.

### Participants

**Study 2a.** Participants were 535 Nigerian adults living in Nigeria at the time of the experiment, and who were recruited via email from contact lists of the authors. Two hundred and sixty-five participants were excluded from the study because they did not watch the experiment videos completely. Participants' video completion rates were assessed by the number of minutes they spent on the experiment page that had the history video embedded. The history videos were the same videos employed in Study 1. The positive history video length was 4:49 minutes and the negative history video length was 5:00 minutes. Hence, participants who were 10 seconds under these video lengths were excluded from the study. In most cases, the excluded participants were 50 to 100+ seconds under the video lengths for their respective conditions. These 265 participants were not paid for the study and were contacted individually via email to inform them about their exclusion and that their data was not going to be used for the study. Therefore, their data were not analysed in this study and are not available in the dataset. This left a final sample size of 270 participants, who were remunerated 1000 Naira mobile phone top-ups for their participation. A sensitivity analysis using G*power 3.1 indicated that the final sample of 270 provides 80% power in the current design ($\alpha = .05$; $df_{num} = 1$) to detect an effect as small as Cohen's $f^2 = 0.17$ (equivalent to $\eta_p^2$ of .029) in a two-way analysis of variance (ANOVA). Our recruitment strategy was to maximise sample size given the limited time available for data collection (data were collected between the 1st of June–the 31st of October of 2020). All participants were aged 17 to 43 ($M = 23.76$, $SD = 5.70$), and included 150 females and 120 males.

**Study 2b.** Participants were 259 Nigerian adults who studied psychology at the Alex Ekwueme Federal University, Ndufu-Alike Ikwo, Abakaliki, Ebonyi State, Nigeria at the time of the experiment. A sensitivity analysis using G*power 3.1 indicated that the final sample of 259 provides 80% power in the current design ($\alpha = .05$; $df_{num} = 1$) to detect an effect as small as Cohen's $f^2 = 0.17$ (equivalent to $\eta_p^2$ of .029) in a two-way analysis of variance (ANOVA). Our strategy was to recruit as many students as possible during a two-day data collection period (11th– 12th of February 2021). Two hundred and seventeen participants were invited to participate in the experiment in a large lecture hall. On arrival at the lecture hall, participants were given the experiment link to complete on their electronic gadgets (i.e., phones, tablets, or laptops). A further 42 participants completed the survey in their accommodation. Participants were not remunerated for participation in this study. All participants were aged 17 to 30 years ($M = 21.76$, $SD = 2.59$). There were 134 females and 118 males, while one identified as 'other' and six did not report their gender. Furthermore, in this study, there were no exclusions from the sample as all participants watched the experimental videos completely.

### Design

The independent variables in both studies were the valence of African history with two levels: positive precolonial history and negative precolonial history; and the subjective importance of

African history with two levels: important history norm and unimportant history norm. Therefore, each study had a 2 (subjective importance of African history: important history norm vs. unimportant history norm) X 2 (historical representation: positive precolonial history vs. negative precolonial history) between-participant design. The dependent variables were collective history appraisals (i.e., subjective importance and valence), in-group (i.e., African) identification, and group-based actions, consisting of consciousness-raising, collective political action, and social competition.

## Materials

All materials as presented to participants can be found on the project OSF site at https://osf.io/qm8g5/?view_only=35292d860cb94550a52d42b01733f87e. Unless otherwise indicated, responses were made on scales from 1 (*strongly disagree*) to 7 (*strongly agree*).

**Subjective importance of African history.** The manipulations were operationalised by way of a fabricated online news article from Afrobarometer (e.g., https://afrobarometer.org/publications/ad512-perceptions-are-bad-reality-worse-citizens-report-widespread-predation-african) reporting on findings of research on Africans' views on African history. In each case, the stimulus article was identical across conditions apart from small wording changes that emphasised that Africans either appraised African history as important or unimportant.

*Important history norm*. The news article's headline read that 'African history is important to Africans, Afrobarometer survey finds'. The first paragraph read that 'A new analysis from Afrobarometer shows that Africans believe that the history of Africa itself is largely relevant to how they, and other Africans, see themselves and the world today'. The second paragraph portrayed that Africans viewed African history as important to their identity, and how they viewed Africa and its future. The last paragraph reported that Africans viewed African history as important to the world more generally.

*Unimportant history norm*. The news article's headline read that 'African history is not important to Africans, Afrobarometer survey finds'. Moreover, the first paragraph read that 'A new analysis from Afrobarometer shows that Africans believe that the history of Africa itself is largely irrelevant to how they, and other Africans, see themselves and the world today'. The second paragraph portrayed that Africans viewed African history as unimportant to their identity, and how they viewed Africa and its future. The last paragraph reported that Africans viewed African history as unimportant to the world more generally.

**Reflection on the subjective importance of African history: 'Three reasons' task.** In order to reinforce and increase participants' engagement with the content of the purported Afrobarometer articles–similar to and/or adapted from Haslam and colleagues' [67] 'three things' manipulation of social identity salience–participants were asked to reflect on and write down three reasons each as to why they think African history is important (importance condition) or unimportant (unimportance condition) to (a) other Africans, (b) the world, and (c) themselves.

**Historical representations.** The same video materials for positive precolonial Africa (which represented African history in positive terms) and negative precolonial Africa (which represented African history in negative terms) that were used in Study 1, were used respectively for the positive precolonial history and negative precolonial history conditions in Studies 2a and 2b.

**Collective history appraisals.** Measures were derived from a separate study that confirmed the same three-factor structure that was obtained from Study 1 for the appraisals of the collective history scale (see confirmatory factor analysis and fit indices in the document 'CFA

for appraisals of collective history scale_ Studies 2a & 2b' on the project OSF site at: https://osf. io/qm8g5/?view_only=35292d860cb94550a52d42b01733f87e).

*Subjective importance.* Six items (Study 2a α = .83; Study 2b α = .66) were used to assess participants' appraisals of the subjective importance of African history (e.g., 'The history of Africa is relevant to the present day' and 'The history of Africa is very rich').

*Valence.* Three items (Study 2a α = .78; Study 2b α = .75) were used to assess participants' appraisals of the valence of African history (i.e., historical representations) (e.g., 'Africa has a glorious history' and 'It is enjoyable thinking about African history').

**Identification.** The four-item measure (Study 2a α = .81; Study 2b α = .82) of social identification as suggested by Postmes and colleagues [68] was used to assess participants' African identification (e.g., 'I identify with Africans' and 'Being African is an important part to how I see myself').

**Group-based action.** *Consciousness-raising.* Six items (Study 2a α = .87; Study 2b α = .85) were used to assess participants' willingness to engage in activities that would raise awareness about African history (e.g., 'Volunteer for an organisation that informs Africans about African history' and 'Learn about and share African history with other Africans'). The response scale ranged from 1 (*very unwilling*) to 7 (*very willing*).

*Collective political action.* The same 9-item scale used in Study 1 was used for Studies 2a (α = .87) and 2b (α = .87).

*Social competition.* The same 4-item scale used in Study 1 was used for Studies 2a (α = .75) and 2b (α = .87).

**Additional measures.** The study contained some additional measures that were included for exploratory reasons and whose data are not analysed here. These were (a) scales that were developed for this research, which included a three-item scale of appraisals of clarity of the in-group's collective history and a three-item scale of appraisals of one's knowledge of the in-group's collective history; and (b) measures that assessed socio-structural factors and were adapted from Mummendey and colleagues [49], which included a single-item measure of the stability of intergroup status relations, a two-item scale of the legitimacy of intergroup status relations, a two-item scale of permeability of group boundaries, and a two-item scale of individual mobility [adapted from 47].

## Procedure

Participants were informed that the study was a questionnaire on the history of Africa and their opinions on Africa, such as how they and other Africans see Africa. First, participants were randomly assigned to read either the article depicting that Africans appraised African history as important or the article that depicted that Africans appraised African history as unimportant. After participants read the articles, they completed the three-reasons task of why African history may be important or unimportant to reinforce the information they read in the respective articles. Next, participants were randomly assigned to either watch a video on positive precolonial or negative precolonial African history. After participants saw the video narratives, they completed the appraisals of collective history scale (). Next, participants completed the measures on identification and group-based action, followed by demographic information (i.e., age, place of birth, African nationality and ethnicity, citizenship, gender, and education level). The same debrief that was given to participants in Study 1 for seeing negative narratives of precolonial Africa was given to participants who were assigned to the negative

**Table 6. Bivariate correlations of all variables in Study 2a.**

| | 1 | 2 | 3 | 4 | 5 |
|---|---|---|---|---|---|
| 1. Subjective importance | 1 | | | | |
| 2. Valence | .40** | 1 | | | |
| 3. Identification | .67** | .29** | 1 | | |
| 4. Consciousness-raising | .37** | .20** | .40** | 1 | |
| 5. Collective political action | .33** | .13** | .35** | .74** | 1 |
| 6. Social competition | .30** | .08 | .32** | .21** | .21** |

Notes.

** $p < .01$

precolonial African history condition in Studies 2a and 2b. Lastly, participants were thanked and debriefed on the purposes, hypotheses, and expected outcomes of the research.

## Results

Bivariate correlations of all variables can be found in Table 6 for Study 2a and Table 7 for Study 2b.

### Collective history appraisals

**Appraised subjective importance of African history.** For Study 2a, a two-way ANOVA with the subjective importance of African history manipulation (important history norm vs unimportant history norm) and historical representation manipulation (positive precolonial history vs negative precolonial history) as factors revealed a significant main effect of the manipulation of the subjective importance of African history on participants' appraised subjective importance of African history, $F(1, 266) = 5.24$, $p = .023$, $\eta_p^2 = .019$. As expected, participants appraised African history as more subjectively important in the important history norm condition ($M = 6.26$, $SD = 1.01$) than in the unimportant history norm condition ($M = 5.93$, $SD = 1.36$). Furthermore, as expected there were null results for the (1) main effect of the manipulation of historical representations on participants' appraised subjective importance of African history, $F(1, 266) = 1.71$, $p =. 192$, $\eta_p^2 = .006$; and (2) interaction effect, $F(1, 266) < 1$. Tables 9 and 10 show the inferential and descriptive statistics respectively for the ANOVA on appraised subjective importance for Study 2a.

**Table 7. Bivariate correlations of all variables in Study 2b.**

| | 1 | 2 | 3 | 4 | 5 |
|---|---|---|---|---|---|
| 1. Subjective importance | 1 | | | | |
| 2. Valence | .41** | 1 | | | |
| 3. Identification | .39** | .29** | 1 | | |
| 4. Consciousness-raising | .40** | .38** | .34** | 1 | |
| 5. Collective political action | .37** | .27** | .31** | .69** | 1 |
| 6. Social competition | .08 | .05 | .16* | .24** | .22** |

Notes.

* $p < .05$

** $p < .01$

For Study 2b, a similar two-way ANOVA revealed a significant main effect of the manipulation of the subjective importance of African history on participants' appraised subjective importance of African history, $F(1, 252) = 9.08$, $p = .003$, $\eta_p^2 = .035$. As expected, participants appraised African history as more subjectively important in the important history norm condition ($M = 6.27$, $SD = 0.94$) than in the unimportant history norm condition ($M = 5.82$, $SD = 1.27$). However, unexpectedly there was also a significant main effect of the manipulation of historical representations on participants' appraised subjective importance of African history, $F(1, 252) = 7.20$, $p = .008$, $\eta_p^2 = .028$, with participants in the positive precolonial history condition ($M = 6.28$, $SD = 1.01$) appraising African history as more subjectively important in comparison to participants in the negative precolonial history condition ($M = 5.87$, $SD = 1.19$). The interaction effect was not significant, $F(1, 252) < 1$. Tables 9 and 10 show the inferential and descriptive statistics respectively for the ANOVA on appraised subjective importance for Study 2b.

**Appraised valence of African history.** For Study 2a, a similar two-way ANOVA was conducted on participants' appraised valence of African history and revealed a significant main effect of the manipulation of historical representations, $F(1, 265) = 61.60$, $p < .001$, $\eta_p^2 = .189$. As expected, participants appraised African history as more positive in the positive precolonial history condition ($M = 5.34$, $SD = 1.49$) than in the negative precolonial history condition ($M = 3.76$, $SD = 1.79$). Furthermore, as expected there were null results for the (1) main effect of the manipulation of the subjective importance of African history on participants' appraised valence of African history, $F(1, 265) < 1$; and (2) interaction effect, $F(1, 265) < 1$. Tables 9 and 10 show the inferential and descriptive statistics respectively for the ANOVA on appraised valence for Study 2a.

For Study 2b, a similar two-way ANOVA was conducted on participants' appraised valence of African history and revealed a significant main effect of the manipulation of historical representations, $F(1, 249) = 30.01$, $p < .001$, $\eta_p^2 = .108$. As expected, participants appraised African history as more positive in the positive precolonial history condition ($M = 5.44$, $SD = 1.57$) than in the negative precolonial history condition ($M = 4.16$, $SD = 2.05$). Furthermore, as expected there were null results for the (1) main effect of the manipulation of the subjective importance of African history on participants' appraised valence of African history, $F(1, 249) < 1$; and (2) interaction effect, $F(1, 249) < 1$. Tables 8 and 9 show the inferential and descriptive statistics respectively for the ANOVA on appraised valence for Study 2b.

**Identification.** For Study 2a, a similar two-way ANOVA was conducted on the scale of identification and revealed null effects of the (1) main effect of the subjective importance of African history on African identification, $F(1, 266) = 3.23$, $p = .073$, $\eta_p^2 = .012$,; (2) main effect of historical representations, $F(1, 266) < 1$; and (3) interaction effect of the subjective importance of African history and historical representations, $F(1, 266) = 1.36$, $p = .245$, $\eta_p^2 = .005$.

**Table 8. Inferential statistics for collective history appraisals ANOVA effects of Studies 2a and 2b.**

| Effect (IV) | DV | Study 2a | | | | Study 2b | | | |
|---|---|---|---|---|---|---|---|---|---|
| | | $F$ | $df$ | $p$ | $\eta_p^2$ | $F$ | $df$ | $p$ | $\eta_p^2$ |
| Subjective importance | Appraised importance | 5.24 | 1, 266 | .023 | .019 | 9.08 | 1, 252 | .003 | .035 |
| | Appraised valence | 0.08 | 1, 265 | .780 | .000 | 0.88 | 1, 249 | .349 | .004 |
| | Appraised importance | 1.71 | 1, 266 | .192 | .006 | 7.20 | 1, 252 | .008 | .028 |
| Valence | Appraised valence | 61.60 | 1, 265 | < .001 | .189 | 30.01 | 1, 249 | < .001 | .108 |
| Subjective importance x Valence | Appraised importance | 0.09 | 1, 266 | .798 | .000 | 0.00 | 1, 252 | .956 | .000 |
| | Appraised valence | 0.13 | 1, 265 | .724 | .000 | 0.16 | 1, 249 | .687 | .001 |

**Table 9. Descriptive statistics for collective history appraisals ANOVA effects of Studies 2a and 2b.**

| DV | Study 2a | | | | Study 2b | | | |
|---|---|---|---|---|---|---|---|---|
| | Subjective importance of African history condition | | | | Subjective importance of African history condition | | | |
| | Important history norm | | Unimportant history norm | | Important history norm | | Unimportant history norm | |
| | M | SD | M | SD | M | SD | M | SD |
| Appraised importance | 6.26 | 1.01 | 5.93 | 1.36 | 6.27 | 0.94 | 5.82 | 1.27 |
| Appraised valence | 4.50 | 1.76 | 4.54 | 1.90 | 4.93 | 1.80 | 4.59 | 2.09 |
| DV | Study 2a | | | | Study 2b | | | |
| | Valence of African history condition | | | | Valence of African history condition | | | |
| | Positive precolonial history | | Negative precolonial history | | Positive precolonial history | | Negative precolonial history | |
| | M | SD | M | SD | M | SD | M | SD |
| Appraised importance | 6.19 | 1.35 | 6.02 | 1.05 | 6.28 | 1.01 | 5.87 | 1.19 |
| Appraised valence | 5.34 | 1.49 | 3.76 | 1.79 | 5.44 | 1.57 | 4.16 | 2.05 |

Tables 10 and 11 show the inferential and descriptive statistics respectively for the ANOVA on identification for Study 2a.

For Study 2b, a similar two-way ANOVA was conducted on the measure of identification and revealed a significant main effect of the subjective importance of African history on African identification, $F(1, 253) = 4.21$, $p = .041$, $\eta_p^2 = .016$ with participants in the important history norm condition ($M = 6.34$, $SD = 1.23$) identifying more with African identity than participants in the unimportant history norm condition ($M = 5.97$, $SD = 1.58$). All other effects were null, these include (1) the main effect of historical representations, $F(1, 253) < 1$; and (2) the interaction effect of the subjective importance of African history and historical representations, $F(1, 253) < 1$. Tables 10 and 11 show the inferential and descriptive statistics respectively for the ANOVA on identification for Study 2b.

**Group-based action.** For Study 2a, a two-way MANOVA with the subjective importance of African history (important history norm vs unimportant history norm) and historical representation (positive precolonial history vs negative precolonial history) as factors was conducted on the group-based action (i.e., consciousness-raising, collective political action, and social competition). Using Wilks' lambda, results revealed null effects of the (1) main effect of the subjective importance of African history, $\lambda = 0.99$, $F(3, 264) = 1.19$, $p = .313$, $\eta_p^2 = .013$; (2) main effect of historical representations, $\lambda = 0.98$, $F(3, 264) = 1.74$, $p = .159$, $\eta_p^2 = .019$; and (3) interaction effect of the subjective importance of African history and historical representations, $\lambda = 0.98$, $F(3, 264) = 1.85$, $p = .138$, $\eta_p^2 = .021$. Table 12 shows the follow-up analyses for the ANOVAs of Study 2a which were consistently null effects. Table 13 shows the descriptive statistics for the ANOVA effects of Study 2a.

For Study 2b, a similar two-way MANOVA revealed (1) a null main effect of the subjective importance of African history, $\lambda = 1.00$, $F(3, 240) = 0.19$, $p = .906$, $\eta_p^2 = .002$; (2) a significant main effect of historical representations, $\lambda = 0.96$, $F(3, 240) = 3.75$, $p = .012$, $\eta_p^2 = .045$ –see below for the accompanying inferential and descriptive statistics from the significant follow-

**Table 10. Inferential statistics for identification ANOVA effects of Studies 2a and 2b.**

| Effect of IV | DV | Study 2a | | | | Study 2b | | | |
|---|---|---|---|---|---|---|---|---|
| | | F | df | P | $\eta_p^2$ | F | df | p | $\eta_p^2$ |
| Subjective importance | Identification | 3.23 | 1, 266 | .073 | .012 | 4.21 | 1, 253 | .041 | .016 |
| Valence | Identification | 0.00 | 1, 266 | .992 | .000 | 0.08 | 1, 253 | .774 | .000 |
| Subjective importance x Valence | Identification | 1.36 | 1, 266 | .245 | .005 | 0.02 | 1, 253 | .894 | .000 |

**Table 11. Descriptive statistics for identification ANOVA effects of Studies 2a and 2b.**

| DV | Study 2a | | | | Study 2b | | | |
|---|---|---|---|---|---|---|---|---|
| | Subjective importance of African history condition | | | | Subjective importance of African history condition | | | |
| | Important history norm | | Unimportant history norm | | Important history norm | | Unimportant history norm | |
| | *M* | *SD* | *M* | *SD* | *M* | *SD* | *M* | *SD* |
| Identification | 6.30 | 0.95 | 6.05 | 1.37 | 6.34 | 1.23 | 5.97 | 1.58 |
| **DV** | **Study 2a** | | | | **Study 2b** | | | |
| | Valence of African history condition | | | | Valence of African history condition | | | |
| | Positive precolonial history | | Negative precolonial history | | Positive precolonial history | | Negative precolonial history | |
| | *M* | *SD* | *M* | *SD* | *M* | *SD* | *M* | *SD* |
| Identification | 6.17 | 1.40 | 6.18 | 0.95 | 6.22 | 1.39 | 6.13 | 1.43 |

up ANOVA main effects of consciousness-raising and collective political action; and (3) a null interaction effect of the subjective importance of African history and historical representations, $\lambda = 1.00$, $F(3, 240) = 0.08$, $p = .974$, $\eta_p^2 = .001$. Table 12 shows the follow-up analyses for the ANOVAs of Study 2b which were consistently null effects apart from significant main effects of historical representations on consciousness-raising [$F(1, 242) = 4.58$, $p = .033$, $\eta_p^2 = .019$, with participants in the positive precolonial history ($M = 6.40$, $SD = 1.08$) condition having higher willingness for consciousness-raising than participants in the negative precolonial history condition ($M = 6.06$, $SD = 1.28$)] and collective political action [$F(1, 242) = 4.21$, $p = .041$, $\eta_p^2 = .017$, with participants in the positive precolonial history ($M = 5.81$, $SD = 1.26$) condition having higher willingness for collective political action than participants in the negative precolonial history condition ($M = 5.45$, $SD = 1.49$)]. Table 13 shows the descriptive statistics for the ANOVA effects of Study 2b.

## Moderated mediation analyses

We sought to directly replicate the moderated mediation analyses from Study 1 of the conditional indirect effect of the manipulation of historical representations (i.e., positive and negative precolonial history in Study 1) on in-group identification and group-based action via appraised/measured valence of African history at different levels of appraised/measured subjective importance of African history (see Fig 1). However, the results for Study 1 were not replicated by the results of Studies 2a or 2b. Specifically, the moderated mediation indexes (*MMI*) were not significant with their 95% confidence intervals including zero for the analysis on (1) identification in Studies 2a (*MMI* = .02, *SE* = .09, 95% *CI* = [-.18, .17]) and 2b (*MMI* = .11, *SE*

**Table 12. Inferential statistics for group-based action follow-up ANOVA effects of Studies 2a and 2b.**

| Effect (IV) | DV | Study 2a | | | | Study 2b | | | |
|---|---|---|---|---|---|---|---|---|
| | | *F* | *df* | *p* | $\eta_p^2$ | *F* | *df* | *p* | $\eta_p^2$ |
| Subjective importance | Consciousness-raising | 0.86 | 1, 266 | .354 | .003 | 0.09 | 1, 242 | .767 | .000 |
| | Collective political action | 1.01 | 1, 266 | .317 | .004 | 0.04 | 1, 242 | .834 | .000 |
| | Social competition | 3.20 | 1, 266 | .075 | .012 | 0.13 | 1, 242 | .723 | .001 |
| Valence | Consciousness-raising | 0.43 | 1, 266 | .513 | .002 | 4.58 | 1, 242 | .033 | .019 |
| | Collective political action | 3.12 | 1, 266 | .078 | .012 | 4.21 | 1, 242 | .041 | .017 |
| | Social competition | 0.56 | 1, 266 | .455 | .002 | 2.45 | 1, 242 | .119 | .010 |
| Subjective importance x Valence | Consciousness-raising | 1.54 | 1, 266 | .216 | .006 | 0.01 | 1, 242 | .944 | .000 |
| | Collective political action | 0.39 | 1, 266 | .535 | .001 | 0.05 | 1, 242 | .817 | .000 |
| | Social competition | 2.80 | 1, 266 | .096 | .010 | 0.19 | 1, 242 | .667 | .001 |

**Table 13. Descriptive statistics for group-based action follow-up ANOVA effects of Studies 2a and 2b.**

| DV | Study 2a | | | | Study 2b | | | |
|---|---|---|---|---|---|---|---|---|
| | Subjective importance of African history condition | | | | Subjective importance of African history condition | | | |
| | Important history norm | | Unimportant history norm | | Important history norm | | Unimportant history norm | |
| | M | SD | M | SD | M | SD | M | SD |
| Consciousness-raising | 6.00 | 1.21 | 5.85 | 1.43 | 6.27 | 1.17 | 6.17 | 1.23 |
| Collective political action | 5.57 | 1.18 | 5.43 | 1.34 | 5.63 | 1.40 | 5.62 | 1.39 |
| Social competition | 6.30 | 1.04 | 6.05 | 1.32 | 6.25 | 1.21 | 6.17 | 1.05 |
| DV | Study 2a | | | | Study 2b | | | |
| | Valence of African history condition | | | | Valence of African history condition | | | |
| | Positive precolonial history | | Negative precolonial history | | Positive precolonial history | | Negative precolonial history | |
| | M | SD | M | SD | M | SD | M | SD |
| Consciousness-raising | 5.98 | 1.37 | 5.88 | 1.27 | 6.40 | 1.08 | 6.06 | 1.28 |
| Collective political action | 5.64 | 1.27 | 5.38 | 1.24 | 5.81 | 1.26 | 5.45 | 1.49 |
| Social competition | 6.24 | 1.10 | 6.12 | 1.29 | 6.32 | 1.02 | 6.09 | 1.23 |

= .06, 95% *CI* = [-.02, .23]); and (2) group-based action in Studies 2a (*MMI* = -.03, *SE* = .09, 95% *CI* = [-.21, .15]) and 2b (*MMI* = .07, *SE* = .05, 95% *CI* = [-.17, .04]). We report all the results of the moderated mediation analyses for Studies 2a and 2b in the supplementary material on the project OSF site: https://osf.io/qm8g5/?view_only= 35292d860cb94550a52d42b01733f87e. Moreover, it is important bear in mind that this attempted replication of the findings of history-as-contrast and history-as-inspiration of Study 1 was made more difficult (or was not a direct replication) due to the various changes in the methodology between Study 1 and Studies 2a and 2b. In other words, direct comparisons were made more difficult due to minor differences in methodology between the studies.

## Discussion

Aiming to replicate the exploratory results of Study 1, in Studies 2a and 2b we orthogonally manipulated the subjective importance of African history (important history norm vs unimportant history norm) and historical representation (positive precolonial history vs negative precolonial history). We predicted that the effect of historical representations–specifically the valence of African history–would also depend on whether African history is appraised as subjectively important, such that there would be an interaction between historical representations and the subjective importance of African history. More precisely, if the effects in Study 1 were reliable, then for participants in the positive precolonial African history condition, the exposure to the condition of subjective importance of African history would lead to higher in-group identification and group-based action in comparison to the condition of subjective unimportance of African history (history-as-inspiration prediction). In contrast, for participants in the negative precolonial African history condition, the exposure to the condition of subjective *unimportance* of African history would also lead to higher in-group identification and group-based action in comparison to the subjective importance condition (history-as-contrast prediction).

Results from Studies 2a and 2b provided no support for this pattern. Most analyses produced null results, with some exceptions (e.g., in Study 2b, there was a significant main effect of the subjective importance of African history on identification, and of historical representations on consciousness-raising and collective political action). Moreover, results from moderated mediation analyses of Studies 2a and 2b using measured appraisals of subjective importance and valence as in Study 1 did not replicate the pattern found in that study. Below,

we consider methodological features that inform the possible interpretation of the null findings obtained.

One possible reason for the null effects in Studies 2a and 2b may be the relative weakness of the manipulations of the subjective importance of African history. Specifically, the effectiveness of the manipulations (i.e., new articles) might have been hampered because they adopted content-free and generic summations of descriptive norms around how Africans appraise the importance of African history. The news articles did not have specific examples of the reasons why Africans may appraise African history as important or unimportant. For example, Africans may appraise African history as important because of Africa's ancient and great civilisations and its contribution to science and technology. Conversely, Africans may appraise African history as unimportant because it is not properly documented and most of the popular African history is about the suffering of African people (e.g., the transatlantic slave trade). Such treatments of the subjective importance of African history may have been more effective than what was adopted in our studies. However, this possible explanation of the null effects does not explain the lack of direct replication for Studies 2a and 2b of the moderated mediation findings suggesting the use of history-as-inspiration and history-as-contrast in Study 1 when considering the measured appraisals of subjective importance.

The findings of some main effects in Study 2b may point to the potential of experimental investigations on the impact of collective history appraisals on in-group engagement. However, these results should in turn be interpreted with caution because they were not discernible in Study 2a. Therefore, we can only conclude that our manipulations in Studies 2a and 2b failed to produce a systematic effect on our dependent variables. This is also the case for the (lack of) moderated mediation effects (observed in Study 1) of the manipulation historical representations on in-group identification and group-based action through appraised/measured valence of African history at different levels of appraised/measured subjective importance of African history, suggesting the use of history-as-inspiration and history-as-contrast.

## General discussion

The present research tested the multi-dimensional nature of how group members appraise their collective history and the impact of those appraisals on in-group identification and group-based action in the African context. In Study 1, we explored the effect of two historical representations (positive precolonial history vs. negative precolonial history) vs. a control (African Savannah). Our expectation for Study 1 was that representations of positive precolonial history (vs. negative precolonial history and African Savannah) would lead to higher in-group identification and group-based action. Results revealed non-significant main effects of historical representations on in-group identification and group-based action. However, the primary aim of this research was to test the multi-dimensional nature of how group members appraised their collective history and the impact of those appraisals on in-group engagement.

We therefore proceeded to explore the conditional indirect effect of historical representation on in-group identification and group-based action via (measured) appraised valence of African history contingent on (measured) appraised subjective importance of African history. Results from this exploratory analysis in Study 1 revealed that (1) for participants in the positive precolonial history condition compared to participants in the negative precolonial history condition, appraising African history as more positive predicted greater African identification and group-based action when that history was also appraised as subjectively important: a possible history-as-inspiration strategy to boost identification and group-based action; and (2) for participants in the negative precolonial history condition compared to participants in the positive precolonial history condition, appraising African history as more negative also led to

greater African identification and group-based action when African history was also appraised as subjectively *unimportant*: a possible history-as-contrast strategy to boost identification and group-based action. This history-as-contrast strategy seems less institutively obvious than the history-as-inspiration strategy because history is normally conceptualised as a tool to assimilate with the present [4, 6, 7, 9, 13, 22, 24, 30]. However, this finding aligns with Makanju et al.'s [62] qualitative work that found in-group engagement was boosted when African history was appraised as positive and unimportant because history was used to orient towards a better future by contrasting with the past. This could be seen as a modernist or futurist approach to in-group engagement, where the group's collective future is believed to be best achieved by explicitly rejecting its past.

Based on the exploratory findings of history-as-inspiration and history-as-contrast in Study 1, Studies 2a and 2b tested the interaction effect between orthogonal manipulations of subjective importance of African history (important history norm vs unimportant history norm) and historical representation (positive precolonial history vs. negative precolonial history). However, results from Studies 2a and 2b did not replicate the findings of Study 1.

Taking the history-as-contrast and history-as-inspiration findings in Study 1 at face value, these exploratory findings may signify that understanding the impact of historical representations on in-group engagement means that we may have to consider its subjective importance as well as its valence. For instance, they may suggest the value of developing a more nuanced understanding of *when* perceived collective continuity is valued by group members, rather than assuming that such continuity is always valued per se [13–15]. It may instead be that sometimes, group members may also want to break away from the collective past, especially if they appraise their in-group's history as negative and unimportant. This may be viewed as a modernistic or future-focused strategy to boost in-group engagement. Furthermore, there is a notable correlation between the appraisals of valence and subjective importance in Study 1, suggesting that these appraisals could affect one another. For example, people who hold negative representations of African history may also deem African history less important. This may further explain the findings of history-as-contrast and history-as-inspiration.

However, the exploratory nature of the history-as-contrast and history-as-inspiration findings in Study 1, together with the failure to replicate them in Studies 2a and 2b, means that we have to be very cautious and speculative in their interpretation. A more appropriate conclusion across all studies may instead be a more general one: that group members call on the in-group's history in different ways to achieve group goals, and that there may be no single, essential relationship between specific historical representations and in-group engagement as previously suggested by previous research that predicts specific effects of positive and negative historical representations on in-group engagement [e.g., 5, 22, 24].

Another theoretical implication from our findings relates to the high average scores of in-group identification and group-based action across the three experiments irrespective of experimental conditions (see analysis on high averages in the document 'Results of one-sample t-tests on identification and group-based action variables for Studies 1, 2a & 2b' on the project OSF site at: https://osf.io/qm8g5/?view_only=35292d860cb94550a52d42b01733f87e). These findings on the face of it go against numerous studies [e.g., 27, 34, 36, 37, 69–71] that suggest that Africans negatively evaluate African identity and are not willing to take progressive action on behalf of the African group or collective (i.e., a collective action problem). Therefore, these high averages may (1) be indicative of African identity being a valued identity that Africans engage with, and (2) contradict the stereotype that Africans are not willing to mobilise or act for the benefit of the in-group. However, we feel such conclusions are in turn premature because most of our participants were university students and were therefore not representative samples of the general African (or even Nigerian) population. Hence, these high averages

may represent the reality that university students are relatively more politicised, social-aware and engaged in political issues. Furthermore, this implication–stemming from the fact that most of our samples were university students–is relevant to all of our findings across the three experiments. Therefore, future research should ensure to have a more representative sample of the general African population.

An additional contribution of the present studies is in demonstrating the potential to effectively vary different appraisal dimensions of collective history in experimental contexts. Across our studies, the historical representations of precolonial African history (i.e., positive precolonial history and negative precolonial history) were effective in manipulating the valence of collective history across all three studies as the magnitude of the effect was large on measured appraisals of valence. Moreover, our manipulation of the descriptive social norms of the subjective importance of collective history in Studies 2a and 2b successfully varied measured appraisals of subjective importance, although it had a small-trivial magnitude of effect. Altogether, our experimental approach to research on collective history appraisal dimensions holds promise for research on the impact of historical representations on in-group engagement. Precisely, experimentation may be a good direction for future research as most of the related research to ours in the African context on collective memory has exclusively employed correlational approaches [e.g., 1] and content analysis of social representations of collective history [e.g., 40, 72–74].

Moreover, the inability of our experimental manipulations to systematically impact dependent variables in Studies 2a and 2b calls into question the effectiveness of adopting one-off [see 75, 76] interventions to change group members' appraisals of their in-group's collective history. Interventions to change group members' appraisals of their in-group's collective history may be more amenable to longitudinal experimental designs where within-person change can be observed over time, instead of our approach of between-person variation. Moreover, approaches to changing other 'big' social perceptions like prejudice also emphasise the importance of cumulative rather than one-off effects [e.g., 77, 78]. This leads to a bigger question of intervention studies in social psychology: there is little theory on the processes involved (e.g., how long it may take) to effectively raise group members awareness and/or positively impact evaluations around key worldview issues such as appraisals of collective history. Future research would do well to investigate and delineate such processes (e.g., psychological stages involved, duration needed to be impactful etc.) in raising consciousness and/or positively impacting worldview evaluations of group members.

## Future research directions

The limitations of this research provide ample future research possibilities. First, our studies only surveyed participants from Nigeria, so our findings may not be generalisable to other African contexts. We examined only Nigerian participants because of pragmatic reasons around the ease of data collection in such a hard-to-reach and under-research African population (e.g., three authors of this research are Nigerian and were able to facilitate data collection among a Nigerian sample). There is some reason to speculate that these processes do generalise: many African countries are in similar stages economically and politically with most classified as 'developing' states and having similar postcolonial challenges [e.g., bad governance, insecurity, dependency on the West, poverty, civil wars; 27, 34]. Nonetheless, future research should aim to investigate other African contexts. For example, we theorised that the legacy of colonialism plays a role in how Africans appraise African history [27, 34] and different African countries had different European colonisers who governed their colonies in varied ways [79, 80]. It is reasonable to expect, for example, possible differences between how Nigerians (who

were colonised by Britain) and Angolans (who were colonised by Portugal) appraise African history and its impact on in-group engagement.

Second, future research could develop more impactful manipulations to shape appraisals of the subjective importance of the in-group's collective history to produce a greater magnitude of effect compared to what was obtained in Studies 2a and 2b. For example, future studies can employ video narratives of group members explaining or discussing why they think the in-group's history is important or unimportant to themselves. Additionally, an option for increasing participants' engagement with experimental manipulations would be to employ a longitudinal experimental design that would be able to capture within-person change regarding appraisals of collective history. Such a longitudinal design would also involve more prolonged, repeated exposure to positive historical representations. For example, participants would have the opportunity to watch full episodes of documentaries on prestigious precolonial Africa [e.g., 28].

Third, future research may also examine the role of the moral typecasting model [81] in the relationship between the valence of African history and the in-group engagement of Africans. Specifically, it could be argued that the positive precolonial history presented in our study portrayed Africans as competent moral agents. As such, a more effective and interesting comparison to this positive precolonial history would be a negative precolonial history that portrays Africans as incompetent moral patients, which aligns with the colonialist narrative of Africa. Such an investigation might distinguish when the valence of African history leads to increased in-group engagement, as Africans may be seen as competent moral agents, versus decreased in-group engagement when they may be seen as incompetent moral patients. Overall, it may be beneficial for such an investigation to explicitly highlight these aspects of the moral typecasting model in the manipulations of valence to better examine the relationship with other dimensions of collective history appraisals, such as subjective importance and clarity.

Fourth, this research programme would benefit from a design that utilises (more engaging) real-life social processes that influence in-group members' appraisals of collective history. Specifically, future research can examine the role of group discussions [e.g., 82] on in-group members' appraisals of their in-group's collective history. Indeed, the group-discussion paradigm has been shown to enhance group identification, belief in group-based action efficacy, and commitment to group-based action [82]. Such a paradigm will be an upgrade to the social process of generic and content-free social norms investigated in this research and would be able to capture (more engaging) real-life social processes involved in changing worldviews and attitudes. For example, examining whether discussing and/or knowing about other in-groupers' views of appraisals of the ingroup's collective history impacts the effect of historical representations.

Fifth, the null findings in Studies 2a and 2b regarding our predictions call into question the findings of Study 1, especially as the results of the (moderated mediation) analyses of Study 1 were not replicated for Studies 2a and 2b. Hence, there may be other variables between the pool of participants across our three experiments that may indicate when collective history may be used as a contrast and an inspiration. For example, there may be an off chance that participants in Study 1 were less aware of the specific historical representations presented to them in comparison to participants in Studies 2a and 2b and that may be able to explain the findings obtained. We give this speculation because narratives of precolonial African history are not popular among Africans [40, 41], and as such the novelty of narratives of precolonial African history may vary between samples of the African population. Therefore, future research may consider and control for, the novelty of historical representations (i.e., positive, and negative precolonial African history) on the conditional indirect effects of manipulations of the valence of collective history on in-group engagement through appraised/measured valence of

collective history at different levels of appraised/measured subjective importance of collective history.

## Conclusion

In conclusion, despite some exploratory findings that are worth further tests in future research, the key implication of the present research is that presenting different narratives regarding the valence of an in-group's collective history may not in itself have a straightforward effect on in-group engagement in the present and future. Instead, the connection between positive and negative representations of an in-group's collective history is likely complicated by other appraisals of that collective history, such as its subjective importance. We in turn need further research to establish reliable effects of different collective history appraisal dimensions in in-group engagement.

## Author Contributions

**Conceptualization:** Damilola Makanju, Andrew G. Livingstone, Joseph Sweetman.

**Data curation:** Damilola Makanju.

**Formal analysis:** Damilola Makanju.

**Investigation:** Damilola Makanju.

**Methodology:** Damilola Makanju, Andrew G. Livingstone, Joseph Sweetman, Chiedozie O. Okafor, Franca Attoh.

**Project administration:** Damilola Makanju.

**Resources:** Damilola Makanju.

**Software:** Damilola Makanju.

**Supervision:** Andrew G. Livingstone, Joseph Sweetman.

**Validation:** Damilola Makanju.

**Visualization:** Damilola Makanju.

**Writing – original draft:** Damilola Makanju.

**Writing – review & editing:** Damilola Makanju, Andrew G. Livingstone, Joseph Sweetman, Chiedozie O. Okafor, Franca Attoh.

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
