## [Decision Letter · Decision Letter 0]

6 May 2024

PONE-D-24-09851How appraisals of an in-group's collective history shape collective identity and action: Evidence in relation to African identityPLOS ONE

Dear Dr. Makanju,

Thank you for submitting your manuscript to PLOS ONE. After careful consideration, we feel that it has merit but does not fully meet PLOS ONE’s publication criteria as it currently stands. Therefore, we invite you to submit a revised version of the manuscript that addresses the points raised during the review process.

We look forward to receiving your revised manuscript.

Kind regards,

Pierre Bouchat

Academic Editor

PLOS ONE

Journal Requirements:

2. Please note that in order to use the direct billing option the corresponding author must be affiliated with the chosen institute. Please either amend your manuscript to change the affiliation or corresponding author, or email us at plosone@plos.org with a request to remove this option.

Additional Editor Comments:

Dear Authors,

I have received three reviews of the manuscript PONE-D-24-09851 "How appraisals of an in-group's collective history shape collective identity and action: Evidence in relation to African identity". I also read your paper carefully and independently, before looking at the reviews. I want to thank you for submitting your work to PLOS ONE and the three reviewers, each of whom is expert in this area, for their service to the field.

As you'll see from reading the reviewers' comments, we all find the document interesting and worthy of praise. At the same time, the comments contain insightful suggestions on how to improve the manuscript. My own reading of the work leads me to agree with this general assessment. That's why I'd like to suggest that you revise the document before it is finally accepted. It is particularly important to include in your cover letter a detailed explanation of your decisions regarding how you have dealt with these comments.

Reviewers' comments:

Reviewer's Responses to Questions

**Comments to the Author**

1. Is the manuscript technically sound, and do the data support the conclusions?

Reviewer #1: Partly

Reviewer #2: Yes

Reviewer #3: Yes

2. Has the statistical analysis been performed appropriately and rigorously? 

Reviewer #1: Yes

Reviewer #2: Yes

Reviewer #3: Yes

3. Have the authors made all data underlying the findings in their manuscript fully available?

Reviewer #1: Yes

Reviewer #2: Yes

Reviewer #3: Yes

4. Is the manuscript presented in an intelligible fashion and written in standard English?

Reviewer #1: Yes

Reviewer #2: Yes

Reviewer #3: Yes

5. Review Comments to the Author

Reviewer #1: It is a very interesting work, with data from a cultural region where there are few studies and from one of the great countries of Africa.

They should review and integrate more articles on social and national identity and collective memory, not only focuse on weird nations,for example those of Rosa Cabecinhas (author cited above) on Lusophone Africa. see also10.18800/psico.201001.005 and see this monograph doi 10.5964/jspp.v5i2.895 and thios one doi.org/10.1111/ajsp.12376

Also the recent ones by JH Liu and colleagues on collective memory.

It is central for a tru open science perespective to review not only mainstream anglophone authors

less i have forwarded to articles on africa, asia and latin america

Manipulation has limited effects and the moderate mediation is not replicatedThe transparency of the authors should be reinforced in my opinion and the article should be published The hypothetical deductive model is a myth and I suggest that the authors better explore their data.

First, since African social identity does not change, it can be used as a dispositional variable.

Previous studies have shown that high social identity leads to minimizing or rejecting negative information - for a relationship between collective culture and national identity see Klein.

Looking for previous studies and relying on them I would contract this effect - I suggest using tertiles because Klein found a curvilinear effect. see doi 10.1111/j.2044-8309.2011.02028.x

Second, I suggest to do a meta-analytic integration of the three studies and provide an estimate of the relationship between identity and valence and the other variables. You can use fixed model and this can be accomplished with excel no nedd of CMA o R

Third, use and aggregate the African collective esteem scale and see if it is identity and collective esteem that predict collective action using multiple regression or SEM.

Of course also you can explore what is specifically related to a positive valence of african past (collective identity or esteem) see 10.1017/S1138741600002456

Fourth, make the scale of collective action more explicit, what is the we (Nigeria, Africa...) what are the claims and what is the enemy of this action.

Minor points are said valance instead of valence

A table is written colact 9 and it is 8

Figures have explanatory paragraphs in bold type and this is a mistake

Reviewer #2: Dear Dr Bouchat,

I have now reviewed the manuscript entitled “How appraisals of an in-group’s collective history shape collective identity and action: Evidence in relation to Africa” and my general thought is that it is an interesting, well-written and well-analysed paper. I do have some minor comments and suggestions, however nothing substantial.

Here are some general comments, mostly in order of appearance.

1. Throughout the text, there are some hard-to-read sentences, often because they are too long. This is already the case with the very first sentence, especially the part “how group members engage with their in-groups in terms of how much they identify with the in-group”.

2. On p. 6, the authors propose a multidimensional approach based on thirteen dimensions, themselves clustered under four superordinate dimensions. This seems like an important conceptual framework that then disappears from the discussion. A discussion about what their factor analysis reveals in relation to their conceptual framework would be worth having. For instance, why did the richness dimension end up loading (after rotation) on the importance factor? Also, I was wondering why they did not use the same items for Study 2A and 2B?

3. On p. 13, the format of the material is not clear. Even though the authors talk about “documentaries”, it could be either written, audio or video documentaries. I would suggest the authors describe the format already in the introduction.

4. In experiment 1, the control condition ends up being a not very good “control”, as it receives the highest scores on all collective historical appraisal dimensions. But more importantly, it is also not different from some of the precolonial history conditions on certain levels (for example it is as positive as the positive precolonial history). Once these manipulation checks revealed this issue, it might have been better to remove it from the analyses and focus on the comparison of interest, which is the two types of precolonial history and their influence on identification and group-based action. Indeed, an ANOVA including the three conditions might not reveal underlying differences between the two historical conditions on the count of this third “in-between” control condition.

5. On p. 23 and 49, the authors compare participants’ scores to the midpoint of their scales. To be honest I do not understand the rationale and do not believe it to be relevant. Is there a theoretical (or an empirical) assumption that the population would be centred on the mean point? If yes, this needs to be discussed before. But in many countries, I would not expect measures of identity to be centred around the midpoint. I would prefer a section on descriptive data at the beginning of each result section describing the scores.

6. On p. 25 (and across study 1 in general), the authors seem to consider their manipulation as a manipulation of the appraisal of subjective importance. However, they proposed similar documentaries, with simply a change in valence. It is not clear to me how this change in valence should also be considered a manipulation of the perceived importance.

7. On p. 31, I would have like the authors to reflect on why the positive history manipulation led to a smaller rating of importance than the negative history manipulation. Furthermore, in the paragraph about the “control” condition (which is not really a “control”), I would have liked to see a discussion about how these results affected their further analyses (not necessary if the authors take on board my suggestion on point 4 and remove the condition for their analyses).

8. For studies 2b, students were enrolled in which course? Psychology?

9. On p. 33, the authors say that they changed their measures of appraisals from semantic differential to attitude statements, but they do not say why. It makes comparisons between studies difficult.

10. On p. 52, the authors propose that one possible reason for the null effects may be the relative weakness of the manipulation of the subjective importance of African history. However, in study 1, there was no manipulation of the importance, and yet they found it moderated the effect of valence. Furthermore, the manipulation did work in both studies 2a and 2b. In other words, no manipulation (study 1) revealed an effect and manipulation (studies 2) revealed no effect. I think this needs to be discussed a bit further.

11. At the beginning of the general discussion on p. 53, the authors indicate finding null effects in experiment 1. I would have liked some reflection as to why this was the case (or at least to do it in the discussion of that study).

Minor details.

- p. 5 valance should be “valence”

- p. 25, first line. The partial eta square should be .016, not .015.

- p. 26-27. The parenthesis “(and not the African Savannah condition)” should be placed earlier in the sentence as it reads as if it the comparison between negative history and Savannah and not positive history and Savannah (there is a similar problem on p. 28).

Reviewer #3: This paper aims at investigating the impact of African history appraisals on Nigerian participants’ social identification and group-based action. It presents three experiments in which versions of Africa’s past were manipulated. One of the strong points of this paper is the use of – rather large – Nigerian samples, given the scarcity of research with African participants. In addition, the research question is original and, in my view, highly relevant. The theoretical intro is appropriate, too. The confirmation (based on a former qualitative study) of a typology of appraisals of history, alone, is an important contribution. Results do not meet all the expectations as studies 2 and 3 failed to replicate the interesting pattern found in Study 1, but these results are presented and discussed in a very honest and transparent manner. Globally, I believe this paper deserves to be published. However, I'd like to highlight a few critical points which, once considered, could make it even better.

- A question that remains open throughout the paper is what exactly was manipulated through these excerpts of videos. They are presented as either positive, negative, or neutral. But the nature of “positive” or negative” should be better explained, in my view. Positive could refer to a dimension of morality, but also of agency, competence, etc. If we refer to Gray & Wegner’s moral typecasting model (2009), it seems that Africa is depicted as agentic and moral in the positive video, whereas it is depicted as a negative moral agent in the negative one? The link with social identification seems relevant since people are probably reluctant to identify with an immoral ingroup, but perhaps less so for collective mobilization because the difference in agency between these two experimental conditions is far from obvious. A comparison between a positive agent and negative patient (i.e. victim) moral typecasts would have made sense. I think the paper would benefit from some more explanations about this, and from a deeper discussion.

- I think it is inappropriate to present measures of historical appraisal as manipulation checks. On the one hand, they seem too remote from the actual manipulations to be presented as such. And if they are used to check whether a manipulation was effective or not, there should be a conclusion about this effectiveness, which is missing. On the other hand, manipulation control measures cannot be used to test hypotheses. Thus, in this paper, valence cannot be presented as a manipulation check and then used as a mediator.

- 25. The decision to use subjective importance as a moderator is justified on the basis that there was no strong effect of the manipulation on this variable. I think a theoretical explanation is also needed.

- 26. As I understand, only one contrast was entered into the analysis (comparing the positive to the other two conditions). I’m not an expert in contrast analyses but I think one should always enter all the contrasts (two contrasts for 3 conditions), either polynomial or Elmert. And it would make more sense to compare the two experimental conditions (linear contrast) and the two experimental conditions to the control condition (quadratic contrast). If there is a justification to do otherwise, it should be better explained.

- The positive correlation between subjective importance and valence should be considered in the interpretations. People who hold negative representations of African history may lower its importance.

- I don’t understand clearly why Studies 2a and 2b were presented as distinct studies since it seems to be the same design, same material used for manipulation, same measures, same kind of population (Nigerian adults), etc. Combining the two samples would increase statistical power and simplify the presentation of results. Alternatively, this choice could be better justified.

- In Discussion, I think the important finding that participants who thought African history was not important seemed to use it to contrast their identity with it deserves some more explanation (it seems less intuitively obvious than the assimilation effect found in the positive condition when history was considered important).

- 55. I found the discussion of non-expected results very honest, cautious and nuanced.

In short, I really enjoyed reading this article and found many aspects to like in it. However, there is some room for improvement. I hope my comments will help the authors in this process. I wish them all the best in pursuing this great line of research!

6. PLOS authors have the option to publish the peer review history of their article (what does this mean?). If published, this will include your full peer review and any attached files.

Reviewer #1: No

Reviewer #2: No

Reviewer #3: No

---

## [Author Response · Author response to Decision Letter 0]

17 Jul 2024

Many thanks for your positive, encouraging response to the initial version of this paper, and we hope that you agree that acting on them has further improved the manuscript.

The main revisions include (1) the addition of an ethic statement subsection in the Method section across studies on pages 11 and 37 – 38; (2) the addition of extra citations to the Introduction based on Reviewer 1’s suggestions; (3) the correction of the spelling of valence across the manuscript and correction of the labelling of the variable key for collective action (colact) in the correlation table on page 20; (4) the breaking up of longer sentences in the manuscript to improve the readability and clarity; (5) the addition of a paragraph discussing the results of the exploratory factor analysis (EFA) on the appraisals of collective history items on page 32; (6) improving the clarity of the rationale for changing the semantic differential collective history appraisals items used in Study 1 to attitude statements items employed in Studies 2a and 2b on page 36; (7) making the documentary format clearer in the description of the manipulations of Study 1 on pages 13 – 14; (8) the removal of the one-sample t-tests from the manuscript and the inclusion of a link to point readers to the results of the one-sample t-tests in the General Discussion where we draw theoretical implications of the high averages of African in-group engagement on page 61; (9) making clearer (with additional words and sentences) the focus of the exploratory analysis on pages 25 – 31; (10) the addition of words to direct readers to the project OSF site for the results of the pairwise comparisons of the AVOVA effects of historical representation on variables of identification and group-based action on pages 23 – 24; (11) the addition of the course that participants were enrolled in for Study 2b on page 39; (12) the addition of an acknowledgement of the difficulty of trying to replicate the exploratory findings of Study 1 for Studies 2a and 2b on page 56; (13) the correction of the partial eta squared on page 26 to 0.16; (14) the inclusion of a paragraph capturing the moral typecasting model (Gray & Wagner, 2009) in the future research directions in the General discussion on pages 64 – 65; (15) the dropping of the name ‘manipulation check’ for the collective history appraisals items and conceptualising them as dependent variables; (16) the inclusion of a theoretical rationale on page 26 for employing valence as a mediator and subjective importance as a moderator; (17) the addition of sentences in the General Discussion on pages 60 – 61 to suggestions that the appraisals of valence and subjective importance may affect one another (because of the fairly notable correlation between these variables); (18) the addition of a detailed rationale on why we decided to separate the analysis of Studies 2a and 2b (and ultimately present the studies as separate studies) in a paragraph on pages 36 – 37; and (19) the addition of a discussion of the history-as-contrast finding in the General Discussion on pages 59 – 60. 

As requested, we have responded to each comment and suggestion of the reviewers in bullet points below each comment/suggestion. The reviewers’ comments are in italics, and the page numbers we refer to are the page numbers in the revised manuscript with Track Changes (i.e., in the track changes view). We hope that this adequately addresses concerns raised regarding the previous version of the manuscript.

Editor’s Comments 

Journal Requirements:

1. Please ensure that your manuscript meets PLOS ONE's style requirements, including those for file naming 

• We have ensured that all PLOS ONE’s style requirements have been met for this resubmission/revision.

2. Please note that in order to use the direct billing option the corresponding author must be affiliated with the chosen institute. Please either amend your manuscript to change the affiliation or corresponding author, or email us at plosone@plos.org with a request to remove this option.

• The affiliation has now been changed to the University of Exeter (UoE) which will be billed for this publication as all the research work was conducted at UoE.

• The ethics statement has now been moved to the method sections, with it now being the first subsection of the method sections (on pages 11 and 37 – 38).

• We have now included all citations/references in the reference list.

Dear Authors,

I have received three reviews of the manuscript PONE-D-24-09851 "How appraisals of an in-group's collective history shape collective identity and action: Evidence in relation to African identity". I also read your paper carefully and independently, before looking at the reviews. I want to thank you for submitting your work to PLOS ONE and the three reviewers, each of whom is expert in this area, for their service to the field.

As you'll see from reading the reviewers' comments, we all find the document interesting and worthy of praise. At the same time, the comments contain insightful suggestions on how to improve the manuscript. My own reading of the work leads me to agree with this general assessment. That's why I'd like to suggest that you revise the document before it is finally accepted. It is particularly important to include in your cover letter a detailed explanation of your decisions regarding how you have dealt with these comments.

• Many thanks for your positive, encouraging response to the initial version of this paper, and we hope that you agree that acting on them has further improved the manuscript. As requested, we have responded to each comment and suggestion of the reviewers in bullet points below each comment/suggestion. The reviewers’ comments are in italics, and the page numbers we refer to are the page numbers in the revised manuscript with Track Changes (i.e., in the track changes view). We hope that this adequately addresses concerns raised regarding the previous version of the manuscript.

Reviewers’ Comments 

Reviewer 1

It is a very interesting work, with data from a cultural region where there are few studies and from one of the great countries of Africa.

• We thank you for the positive assessments of our work.

They should review and integrate more articles on social and national identity and collective memory, not only focuse on weird nations,for example those of Rosa Cabecinhas (author cited above) on Lusophone Africa. see also10.18800/psico.201001.005 and see this monograph doi 10.5964/jspp.v5i2.895 and thios one doi.org/10.1111/ajsp.12376

Also the recent ones by JH Liu and colleagues on collective memory.

It is central for a tru open science perespective to review not only mainstream anglophone authorsless i have forwarded to articles on africa, asia and latin America

• Thank you for your suggestions which have now been included in the citations in the introduction, and add to the wide range of non-WEIRD research already cited, such as Makanju et al., 2020, 2023; Licata et al.,2018 (of which Cabecinhas and Liu JH were co-authors); Cabecinhas et al., 2011; Klein & Licata, 2003; Bulhan, 1985b, 2015; Rodney, 2012; Fanon, 1967; Figueiredo et al., 2018; Khan, Svensson, Jogdand & Liu, 2017; Liu et al., 1999; etc.

• More precisely, we have added (1) the citations of Figueiredo et al., 2017 and Ionescu et al., 2021 (citations 11 and 12) to the ‘Historical representations and social identities’ subsection of the Introduction; and (2) the citations of Rottenbacher & Espinosa, 2010 and Roth et al., 2017 (citations 23 & 24) to the first paragraph of the ‘Appraising the in-group’s collective history’ subsection of the Introduction.

Manipulation has limited effects and the moderate mediation is not replicatedThe transparency of the authors should be reinforced in my opinion and the article should be published

• Thank you for your positive assessment of our manuscript.

The hypothetical deductive model is a myth and I suggest that the authors better explore their data.

• It was a little unclear to us whether the reviewer was referring to the model in the paper or the more general hypothetico-deductive paradigm of research. In either case, the paper does include a combination of confirmatory and exploratory analyses, including much more transparent exploration than is typically reported in experimental studies.

First, since African social identity does not change, it can be used as a dispositional variable.

Previous studies have shown that high social identity leads to minimizing or rejecting negative information - for a relationship between collective culture and national identity see Klein.

Looking for previous studies and relying on them I would contract this effect - I suggest using tertiles because Klein found a curvilinear effect. see doi 10.1111/j.2044-8309.2011.02028.x

• Indeed, in our exploration of our data, we have treated African identity as a dispositional variable and we have consistently found that it does not moderate any effects. Ultimately, our data is openly available on the project OSF site, so the reviewer is welcome to explore them in ways not covered in the paper.

Second, I suggest to do a meta-analytic integration of the three studies and provide an estimate of the relationship between identity and valence and the other variables. You can use fixed model and this can be accomplished with excel no nedd of CMA o R

• We appreciate this suggestion. However, we do feel that the current analysis fits the purpose of understanding the multi-dimensional nature of collective history appraisals and their relationship with in-group engagement. More importantly, meta-analytic syntheses across original studies are most useful when (1) there are non-zero effects, and (2) there is some variation in those effects, leaving some ambiguity about overall inferences. In the present case, the inference is clear: the manipulations do not have reliable direct effects on any of the ingroup engagement outcomes. A meta-analyses of these hypothesised effects would therefore almost certainly reconfirm this point, and thus be redundant. Therefore, we will not be doing any additional analysis. Ultimately, our data is available on the Project OSF site and we hope other individuals can engage with them in different and useful ways.

Third, use and aggregate the African collective esteem scale and see if it is identity and collective esteem that predict collective action using multiple regression or SEM.

Of course also you can explore what is specifically related to a positive valence of african past (collective identity or esteem) see 10.1017/S1138741600002456

• We thank you for this suggestion. However, we did not follow the specific rationale for this analysis, and are hesitant to add unfocused exploratory analyses to the already lengthy paper. If the reviewer is curious about specific analyses that we do not currently report, our data are available on the Project OSF site and we hope other individuals can engage with them in different and useful ways.

Fourth, make the scale of collective action more explicit, what is the we (Nigeria, Africa...) what are the claims and what is the enemy of this action.

• We have copies of the questionnaires of the studies on the project OSF site at OSF | How appraisals of an in-group's history shape collective identity and action: Evidence in relation to African identity. Our studies were on the African identity (and not the Nigerian identity) and all the collective action items had clear and explicit targets. For example, the items on consciousness-raising were directed at making other Africans aware of African history. As another example, the items on social competition were directed at Africans competing with the West. Additionally, the items on collective action are explained adequately in the methodology sections of the studies. 

Minor points are said valance instead of valence

A table is written colact 9 and it is 8

Figures have explanatory paragraphs in bold type and this is a mistake

• Thank you for pointing these out. We have now corrected these.

Reviewer 2

I have now reviewed the manuscript entitled “How appraisals of an in-group’s collective history shape collective identity and action: Evidence in relation to Africa” and my general thought is that it is an interesting, well-written and well-analysed paper. I do have some minor comments and suggestions, however nothing substantial.

• We thank you for the positive feedback on our manuscript and the outcome of the review.

Here are some general comments, mostly in order of appearance.

1. Throughout the text, there are some hard-to-read sentences, often because they are too long. This is already the case with the very first sentence, especially the part “how group members engage with their in-groups in terms of how much they identify with the in-group”.

• Thank you for this comment. We have reviewed the manuscript to break up longer sentences to improve the readability and clarity (including the sentence mentioned).

2. On p. 6, the authors propose a multidimensional approach based on thirteen dimensions, themselves clustered under four superordinate dimensions. This seems like an important conceptual framework that then disappears from the discussion. A discussion about what their factor analysis reveals in relation to their conceptual framework would be worth having. For instance, why did the richness dimension end up loading (after rotation) on the importance factor? Also, I was wondering why they did not use the same items for Study 2A and 2B?

• We have now included a paragraph discussing the results of the exploratory factor analysis (EFA) on the appraisals of collective history items on page 32. Specifically, we discuss what the results means with the richness (consisting of depth, temporalty and existence) superordinate dimension loading unto the subjective importance dimension. 

• This present research was part of a body of research of the first author’s doctoral thesis. Therefore, at the start of Studies 2a and 2b, a separate quantitative study had been completed on the scale confirmation of appraisals of collective history items (employing a confirmatory factor analysis - CFA). Consequently, we thought it made sense to use the confirmed scale items for subsequent work on the thesis, especially as the factor structure (of the CFA) matched that of the EFA results in Study 1. Moreover, the confirmed scale items were attitude statement items. Hence, we reasoned that in most social psychological research the attitude statement format is used instead of the semantic differential format (employed in Study 1), so we thought this was an additional advantage in using the confirmed scale items – to follow the norm of most studies in social psychological research. We have now made this rationale clearer by adding additional words on page 36 where we introduce the changes of measures to the appraisals of collective history. This is in addition to what is on page 41 under the materials section in the ‘Collective history appraisals’ subsection which also points out this change. 

3. On p. 13, the format of the material is not clear. Even though the authors talk

---

## [Editor Report · Decision Letter 1]

30 Jul 2024

How appraisals of an in-group's collective history shape collective identity and action: Evidence in relation to African identity

PONE-D-24-09851R1

Dear Dr. Makanju,

Thank you for your responses to the reviewers' comments. After careful reading of these responses and the current version of the manuscript, I find it ready for acceptance in Plos. I believe that your manuscript will make a valuable contribution to our field and thank you for that.

Kind regards,

Pierre Bouchat

Academic Editor

PLOS ONE

---

## [Editor Report · Acceptance letter]

29 Aug 2024

PONE-D-24-09851R1 

PLOS ONE

Dear Dr. Makanju, 

I'm pleased to inform you that your manuscript has been deemed suitable for publication in PLOS ONE. Congratulations! Your manuscript is now being handed over to our production team.

Kind regards, 

on behalf of

Dr. Pierre Bouchat 

Academic Editor

PLOS ONE